# Towards Universal Robust Federated Learning via Meta Stackelberg Game

## ABSTRACT

Recent studies have revealed that federated learning (FL) systems are susceptible to a range of security threats. Although various defense mechanisms have been proposed, they are typically non-adaptive and tailored to specific types of attacks, leaving them insufficient in the face of unknown/uncertain or adaptive attacks. In this work, we formulate adversarial federated learning as a Bayesian Stackelberg Markov game (BSMG) to tackle adaptive attacks of uncertain types. We further develop an efficient meta-learning approach to solve the game, which provides a robust and adaptive FL defense. Theoretically, we show that our algorithm provably converges to the first-order $\varepsilon$-equilibrium point in $O(\varepsilon^{-2})$ gradient iterations with $O(\varepsilon^{-4})$ samples per iteration. Empirical results show that our meta-Stackelberg framework obtains superb performance against strong model poisoning and backdoor attacks with uncertain types.

## 1 INTRODUCTION

Federated learning (FL) allows multiple devices with private data to jointly train a learning model without sharing their local data (McMahan et al., 2017). However, FL systems are vulnerable to adversarial attacks such as untargeted model poisoning attacks and targeted backdoor attacks. To address these vulnerabilities, various robust aggregation rules such as Krum (Blanchard et al., 2017), coordinate-wise median (Yin et al., 2018), trimmed mean (Yin et al., 2018), and FLTrust (Cao et al., 2021) have been proposed to defend against untargeted attacks. Additionally, various post-training defenses such as Neuron Clipping (Wang et al., 2022) and Pruning (Wu et al., 2020) have been proposed recently to mitigate backdoor attacks.

However, the existing defense mechanisms are plagued by incomplete information in adversarial federated learning, where the defender is unaware of the specific attack methods in the FL process. This incomplete information may render the state-of-the-art specialized defenses ineffective should the actual attacks employ different strategies from the expected, leaving the defender unprepared. A simple example observed in Figure 1 is that a mixture of model poisoning and backdoor attacks can significantly degrade the effectiveness of FLTrust and Neuron Clipping, which are designed for countering the two kinds of attacks, respectively. Another example in Figure 1 is that defense policies, designed for non-adaptive attacks mentioned above, prove inadequate when facing adaptive attacks, such as reinforcement-learning-based attacks (Li et al., 2023). Addressing incomplete information is key to the paradigm shift from specialized defense to universal robustness against a variety of attacks.

Prior works have attempted to tackle this incomplete information through two distinct approaches. The first approach is the "infer-then-counter" approach, where the hidden information regarding the attacks is first inferred through observations. For example, one can infer the backdoor triggers through reverse engineering using model weights (Wang et al., 2019a), based on which the backdoor attacks can be mitigated (Zhao et al., 2021). The inference helps adapt the defense to the present malicious attacks. However, this inference-based adaptation requires prior knowledge of the potential attacks (i.e., backdoor attacks) and does not directly lend itself to mixed/adaptive attacks. Moreover, the inference and adaptation are offline, unable to counter online adaptive backdoor attack Li et al. (2022a). The other approach explored the notion of robustness that prepares the defender for the worst case (Sinha et al., 2018), which often leads to a Stackelberg game (SG) between the defender and the attacker. Considering the incomplete information, Sengupta & Kambhampati (2020) proposes a Bayesian SG model (BSG) to capture the interactions under uncertainty. The resulting Stackelberg equilibrium (SE) defines a defense policy targeting the average of all attack methods, assuming the presence of every possible attack in the FL. Yet, such a Stackelberg approach often leads to conservative defense fixed through the FL, which is less flexible than the "infer-then-counter." Recent

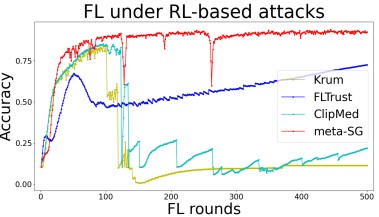 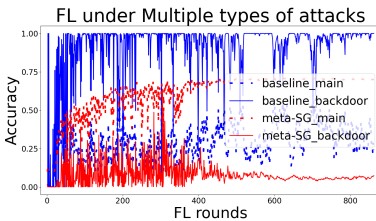

Figure 1: Advantages of the meta-SG framework against the RL-based model poisoning attack Li et al. (2022a) on MNIST with 20% malicious devices (left) and a mix of the backdoor attack against FL (BFL) Bagdasaryan et al. (2020) (5% malicious devices) and the inner product manipulation (IPM) based model poisoning attack Xie et al. (2020) (10% malicious devices) on CIFAR-10 (right). The baseline defense combines the training-stage FLTrust and the post-training Neuron Clipping.

advances in meta-learning (Finn et al., 2017) bring up a data-driven adaptation that tailors a base policy to the testing task using gradient steps. Skipping the inference procedure, meta-learning only requires a handful of samples from the online execution to adapt the policy without prior knowledge. Thanks to its adaptability, the meta-learning defense can outperform the robust one under incomplete information, as observed in (Ge et al., 2023).

Inspired by this data-driven adaptation, this work proposes a novel defense framework integrating the Stackelberg game model with meta-learning, which we refer to as the meta-Stackelbeg game model (meta-SG). Built upon the Stackelberg equilibrium (SE), our meta-SG moves one step further by incorporating the online gradient adaptation into the SE. We refer to this new equilibrium concept as the meta-Stackelberg equilibrium (meta-SE), which offers a computationally efficient data-driven approach to address incomplete information in adversarial FL and enables strategic online adaptation in the presence of various attacks. To the best of our knowledge, this work is among the first endeavors to explore online adaptable defense in FL powered by meta-learning.

Following the meta-learning practice (Finn et al., 2017), the meta-SG framework consists of two stages: pre-training and online adaptation, see Figure 2. The pre-training aims to obtain a base policy (also called meta policy) to be adapted in the second stage. Taking place in an offline simulated environment, the pre-training can be viewed as a Bayesian Stackelberg Markov game (BSMG) between the defender and a set of attacks sampled from the attack domain. To solve the BSMG in the pre-training phase, we propose a meta-Stackelberg learning (meta-SL), a two-timescale policy gradient algorithm, where the policy gradient estimate is Hessian-free due to the strictly competitive nature of BSMG. meta-SL provably converges to the first-order $\varepsilon$-approximate meta-SE in $O(\varepsilon^{-2})$ iterations, and the associated sample complexity per iteration is of $O(\varepsilon^{-4})$. This complexity matches the state-of-the-art results in nonconvex bi-level stochastic optimization (Ji et al., 2021).

Once the game is solved and the equilibrium policy obtained, we move to the online adaptation stage, where the defender starts by using pre-trained policy to interact with the true FL environment while collecting data, such as global model weights and clients' model updates. Then, the defense policy is updated by gradient steps using the data. Of note, the defender is unaware of the actual attacks in the online adaptation phase. Chances are that these attacks may or may not be included in the attack domain in the pre-training. We use notions of uncertain and unknown attacks to distinguish the two cases, respectively. The former refers to those involved in the pre-training stage but undisclosed in the online FL process, leaving the defender unsure about their existence. The latter points to those excluded in the pre-training, to which the defender is never exposed. Thanks to the meta-learning's generalizability (Fallah et al., 2021b), meta-SG gives decent defense performance in both cases.

**Our contributions** are summarized as follows. Due to the space limit, an extended discussion of related work is deferred to Appendix A.

- We address critical security problems in FL with incomplete information on multiple adaptive (non-adaptive) attacks of uncertain/unknown types.
- We develop a Bayesian Stackelberg Makrov game (Section 2.2) to capture the incomplete information in the adversarial FL.
- To equip the defender with strategic adaptability, we propose a new equilibrium concept: meta-Stackelberg equilibrium (Definition 2.1), where the defender (the leader) commits to a meta-learning policy, leading to a data-driven approach to tackle incomplete information.
- To learn the meta equilibrium defense in the pre-training phase, we develop meta-Stackelberg learning (Algorithm 1), an efficient first-order meta RL algorithm, which provably converges to $\varepsilon$-approximate equilibrium in $O(\varepsilon^{-2})$ gradient steps with $O(\varepsilon^{-4})$ samples per iteration, matching the state-of-the-art in stochastic bilevel optimization.

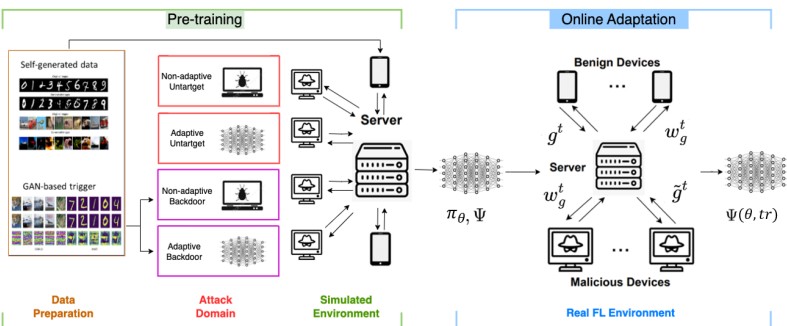

Figure 2: A schematic illustration of the meta-Stagberg game framework. In the pertaining stage, a simulated environment is constructed using generated data and a set of attacks. The defender utilizes meta-Stackelberg learning (Algorithm 1) to obtain the meta policy $\theta$ and the gradient adaptation $\Psi$ in (3). In the online execution, the defender can adapt its defense using gradient steps prescribed by $\Psi(\theta, tr)$ using a sequence of online observations (trajectories) under incomplete information.

- We conduct extensive experiments in real-world settings to demonstrate the superb performance of our proposed method.

## 2 META STACKELBERG DEFENSE FRAMEWORK

### 2.1 FEDERATED LEARNING AND THREAT MODEL

**FL objective.** Consider a learning system that includes one server and $n$ clients, each client possesses its own private dataset $D_i = (x_i^j, y_i^j)_{j=1}^{|D_i|}$ and $|D_i|$ signifies the size of the dataset for the $i$-th client. Let $U = \{D_1, D_2, \ldots, D_n\}$ represent the compilation of all client datasets. The objective of federated learning is defined as identifying a model $w$ that minimizes the average loss across all the devices: $\min_w F(w, U) := \frac{1}{n} \sum_{i=1}^n f(w, D_i)$, where $f(w, D_i) := \frac{1}{|D_i|} \sum_{j=1}^{|D_i|} \ell(w, (x_i^j, y_i^j))$ is the local empirical loss with $\ell(\cdot, \cdot)$ being the loss function.

**Attack objective.** We consider two major categories of attacks, namely, backdoor attacks and untargeted model poisoning attacks. Our framework can be extended to other attack scenarios. For simplicity, assume that the first $M_1$ malicious clients carry out the backdoor attack and the following $M_2$ malicious clients undertake the poisoning attack. The model poisoning attack aims to maximize the average model loss, i.e., $\max_w F(w)$; the backdoor attack aims to preserve decent performance on clean test inputs ("main task") while causing misclassification of poisoned test inputs to one or more target labels ("backdoor task"). Each malicious client in the backdoor attack produces a poisoned data set $D'_{i \leq M_1}$, obtained by altering a subset of data samples $(x_i^j, y_i^j) \in D_i$ to $(\hat{x}_i^j, c^*)$, where $\hat{x}_i^j$ is the tainted sample with a backdoor trigger inserted, and $c^* \neq y_i^j, c^* \in C$ is the targeted label. Let $U' = \{D'_1, D'_2, \ldots, D'_{M_1}\}$ denote the compilation of poisoned datasets. The objective function in the backdoor attack is defined as: $\min_w F'(w) = \lambda F(w, U) + (1 - \lambda) F(w, U')$, where $\lambda \in [0, 1]$ serves to balance between the main task and the backdoor task.

**FL process.** At each round $t$ out of $H$ FL rounds, the server randomly selects a subset of clients $\mathcal{S}^t$ and sends them the most recent global model $w_g^t$. Every benign client in $\mathcal{S}^t$ updates the model using their local data via one or more iterations of stochastic gradient descent and returns the model update $g^t$ to the server. Conversely, an adversary in $\mathcal{S}^t$ creates a malicious model update $\tilde{g}^t$ clandestinely and sends it back. The server then collects the set of model updates $\{\tilde{g}_i^t \cup \tilde{g}_j^t \cup g_k^t\}_{i,j,k \in \mathcal{S}^t, i \in [M_1], j \in [M_2], k \notin [M_1] \cup [M_2]}$, utilizing an aggregation rule $Aggr$ to combine them and updates the global model $w_g^{t+1} = w_g^t - Aggr(\tilde{g}_i^t \cup \tilde{g}_j^t \cup g_k^t)$, which is then sent to clients in round $t + 1$. At the final round $T$, the server applies a post-training defense $h(\cdot)$ on the global model to generate the final global model $\hat{w}_g^T = h(w_g^T)$.

**Attacker type and behavior.** In real FL, multiple types of attacks from various categories may occur simultaneously. For the sake of clarity, we hypothesize a single mastermind attacker present within the FL system who controls a group of malicious clients employing diverse attack strategies, which may be either non-adaptive or adaptive. Non-adaptive attacks involve a fixed attack strategy that solves a short-sighted optimization problem against federated learning system, disregarding the defense mechanism implemented by the server (i.e., the robust aggregation rule and the post-training

defense). Such attacks include inner product manipulation (IPM) (Xie et al., 2020), and local model poisoning attack (LMP) (Fang et al., 2020), federated backdoor attack (BFL) (Bagdasaryan et al., 2020), distributed backdoor attacks (DBA) (Xie et al., 2019), etc. On the other hand, an adaptive attack, such as the RL-based model poisoning attack (Li et al., 2022a) and RL-based backdoor attack (Li et al., 2023), designs model updates by simulating the server's reactions to optimize a long-term objective. One significant hurdle in addressing covert attacks in adversarial settings is incomplete information (Li et al., 2022d), where the server (i.e., the defender) lacks knowledge of the behavior and identities of malicious clients in a realistic black-box scenario. We denote the collective attack configuration of malicious clients as the type of mastermind attacker, detailing $M_1, M_2$, attack behaviors (adaptive or not), and other required parameters of the attack.

## 2.2 BAYESIAN STACKELBERG MARKOV GAME

We model the adversarial FL as a Bayesian Stackelberg Markov game between the defender and the attacker, which is defined by the tuple $G = (\mathcal{P}, Q, S, O, A, \mathcal{T}, r, \gamma)$, where $\gamma \in (0, 1)$ is the reward discounting factor. 1) The player set $\mathcal{P} = \{\mathcal{D}, \mathcal{A}\}$ contains $\mathcal{D}$ as the leader (defender), and $\mathcal{A}$ as the follower (attacker) who controls multiple malicious clients. 2) $Q(\cdot) : \Xi \to [0, 1]$ denotes the probability distribution over the attacker's private types. $\Xi := \{\xi_i\}_{i=1}^{|\Xi|}$ where $\xi_i$ denotes $i$-th type attacks. 3) $S$ is the state space; the state at round $t$ is defined as $s^t := (w_g^t, \mathbf{I}^t)$, where $w_g^t$ is the global model parameters, and $\mathbf{I}^t \in \{0, 1\}^{|\mathcal{S}^t|}$ is the identity vector for the randomly selected clients' subset $\mathcal{S}^t$, where the identities of malicious and benign devices are 1 and 0 respectively. 4) $O$ is the observation space; the observation for the server (i.e., defender) at round $t$ is $w_g^t$ (the server does not have access to the client's identities); the observation for the attacker at round $t$ is $s^t := (w_g^t, \mathbf{I}^t)$ since the attacker controls these malicious clients. 5) $A = \{A_\mathcal{D}, A_\xi\}$ is the joint action set, where $A_\mathcal{D}$ and $A_\xi$ denote the set of defense actions and type-$\xi$ attack actions, respectively; in the FL setting, $a_\mathcal{D}^t = \widehat{w}_g^{t+1} := h(w_g^{t+1})$, and the attacker's action is characterized by the joint actions of malicious clients $a_{A_\xi}^t := \{\widetilde{g}_i^t\}_{i=1}^{M_1} \cup \{\widetilde{g}_i^t\}_{i=M_1+1}^{M_2}$. Note that a malicious device not sampled at round $t$ does not send any information to the server; hence its action has no effect on the model update. The subscript $\xi$ is suppressed if it is clear from the context. 6) $\mathcal{T} : S \times A \to \Delta(S)$ is the state transition, determined by the joint actions and server's subsampling. 7) $r = \{r_\mathcal{D}, r_{A_\xi}\}$, where $r_\mathcal{D} : S \times A \to \mathbb{R}_{\leq 0}$ and $r_{A_\xi} : S \times A \to \mathbb{R}$ are the reward functions for the defender and the attacker, respectively. Define the expected reward at round $t$ as $r_\mathcal{D}^t := -\mathbb{E}[F(\widehat{w}_g^{t+1})]$ and $r_{A_\xi}^t := \rho \mathbb{E}[F'(\widehat{w}_g^{t+1})] - (1-\rho)\mathbb{E}[F(\widehat{w}_g^{t+1})]$, $\rho = M_1/(M_1 + M_2)$, if $\mathbf{1} \cdot \mathbf{I}^t > 0$, and $r_{A_\xi}^t := 0$ otherwise.

In BSMG, the defender first selects the defense policy (the leader), to which the attacker (the follower), randomly drawn from $\Xi$, best responds. This randomness (Bayesian nature) originates from the defender's unawareness of the actual attack type. This best response arises from that the adaptive attacks (Li et al., 2022a; 2023) can learn the optimal attack strategy against the running defense policy, see (2).

## 2.3 META STACKELBERG EQUILIBRIUM

We now articulate the proposed meta-equilibrium, a synthesis of meta-learning and Stackelberg equilibrium to be defined in this subsection. Some helpful notations are introduced below. The defender's and the attacker's policies are parameterized by neural networks $\pi_\mathcal{D}(a_\mathcal{D}^t|s^t; \theta)$, $\pi_\mathcal{A}(a_\mathcal{A}^t|s^t; \phi, \xi)$ with model weights $\theta \in \Theta$ and $\phi \in \Phi$, respectively. Given the two players' policies $\theta, \phi$ and the private attack type $\xi$, the defender's expected utility is defined as $J_\mathcal{D}(\theta, \phi, \xi) := \mathbb{E}_{a_\mathcal{A}^t \sim \pi_\mathcal{A}, a_\mathcal{D}^t \sim \pi_\mathcal{D}}[\sum_{t=1}^H \gamma^t r_\mathcal{D}(s^t, a_\mathcal{D}^t, a_\mathcal{A}^t)]$. Similarly, the attacker's expected utility is $J_\mathcal{A}(\theta, \phi, \xi) := \mathbb{E}_{a_\mathcal{A}^t \sim \pi_\mathcal{A}, a_\mathcal{D}^t \sim \pi_\mathcal{D}}[\sum_{t=1}^H \gamma^t r_\mathcal{A}(s^t, a_\mathcal{D}^t, a_\mathcal{A}^t)]$. Denote by $\tau_\xi := (s^k, a_\mathcal{D}^k, a_\mathcal{A}^k)_{k=1}^H$ the trajectory of the BSMG under type-$\xi$ attacker, which is subject to the distribution $q(\theta, \phi, \xi) := \prod_{t=1}^H \pi_\mathcal{D}(a_\mathcal{D}^t|s^t; \theta)\pi_\mathcal{A}(a_\mathcal{A}^t|s^t; \phi, \xi)\mathcal{T}(s^{t+1}|s^t, a_\mathcal{D}^t, a_\mathcal{A}^t)$. In the later development of meta-SG, we consider the gradient $\nabla_\theta J_\mathcal{D}(\theta, \phi, \xi)$ and its sample estimate $\hat{\nabla}_\theta J_\mathcal{D}(\tau_\xi)$ based on the trajectory $\tau_\xi$. The estimation is due to the policy gradient theorem Sutton et al. (2000) reviewed in Appendix B, and we note that such an estimate takes a batch of $\tau_\xi$ (the batch size is $N_b$) for variance reduction.

To motivate the proposed meta-SE concept, we first present the meta-learning approach and its limitations. Originally proposed for Markov decision processes (MDP) (Finn et al., 2017), meta-learning mainly targets non-adaptive attacks, where $\pi_\mathcal{A}$ is a pre-fixed attack strategy, such as IPM and LMP. In this case, BSMG reduces to a family of MDPs, where transition kernels are dependent

on the type-$\xi$ attack strategy, i.e., $T_\xi(\cdot|s, a_\mathcal{D}) := \int_A T(\cdot|s, a_\mathcal{D}, a_\mathcal{A}) d\pi_A(a_\mathcal{A}|s; \phi, \xi)$. Meta-learning aims to pre-train a base policy on a variety of attacks (i.e., MDPs) from the attack domain such that a one-step gradient adaption applied to the base produces a decent defense against the actual attack in the online environment. Mathematically, the base policy in meta-learning is as below, and the adaptation is given by $\theta + \eta \hat{\nabla}_\theta J_\mathcal{D}(\tau)$. In practice (Nichol et al., 2018; Finn et al., 2017) and our experiments, multi-step gradient adaptation can also be employed, denoted as $\Psi(\theta, \tau)$ for brevity. An extended review on meta-learning is in Appendix B.

$$\max_\theta \mathbb{E}_{\xi \sim Q(\cdot)} \mathbb{E}_{\tau \sim q(\theta)} [J_\mathcal{D}(\theta + \eta \hat{\nabla}_\theta J_\mathcal{D}(\tau), \phi, \xi)] \tag{1}$$

The meta-learning defense fails to account for the adaptive attacker that learns to evade the defense as showcased in (Li et al., 2022a; 2023). The attacker's learning process aims to maximize the attack performance under the running defense, leading to the best response defined in the constraint in (2). Anticipating these intelligent attackers, a rational defender seeks to find the optimal policy that solves the following optimization, leading to a Stackelberg equilibrium (SE) defense.

$$\max_{\theta \in \Theta} \mathbb{E}_{\xi \sim Q(\cdot)} [J_\mathcal{D}(\theta, \phi_\xi^*, \xi)] \quad \text{s.t.} \; \phi_\xi^* \in \arg\max J_\mathcal{A}(\theta, \phi, \xi), \forall \xi \in \Xi. \tag{2}$$

The SE defense targets a "representative" attacker, an average of all attack types, and such a defense is fixed throughout the online execution. Even though such an equilibrium admits a simple characterization, its limitation is also evident: the defender does not adapt to the specific attacker in the online execution. To equip the defender with responsive intelligence under incomplete information, we propose a new equilibrium concept, meta-Stackelberg equilibrium in Definition 2.1.

**Definition 2.1** (Meta Stackelberg Equilibrium). The defender's meta policy $\theta$ and the attacker's type-dependent policy $\phi$ constitute a meta Stackelberg equilibrium if they satisfy

$$\max_{\theta \in \Theta} \mathbb{E}_{\xi \sim Q} \mathbb{E}_{\tau \sim q} [J_\mathcal{D}(\theta + \eta \hat{\nabla}_\theta J_\mathcal{D}(\tau), \phi_\xi^*, \xi)], \text{s.t.} \; \phi_\xi^* \in \arg\max \mathbb{E}_{\tau \sim q} J_\mathcal{A}(\theta + \eta \hat{\nabla}_\theta J_\mathcal{D}(\tau), \phi, \xi). \tag{3}$$

Meta-SE combines the best of two worlds: it creates an adaptable defense anticipating that adaptive attackers would learn to best respond to the adapted policy. In other words, this meta-SE policy $\theta$, learned in pre-training, takes into account the attacker's reaction in the online stage, creating a **strategic adaptation**. This strategic adaptation addresses incomplete information in a data-driven manner, leading to a tractable computation scheme for large-scale FL systems in reality. As a comparison, we review perfect Bayesian equilibrium in Appendix C, a Bayesian-posterior approach to handle incomplete information, which soon becomes intractable as the dimensionality increases.

## 2.4 META STACKELBERG LEARNING AND ONLINE ADAPTATION

The purpose of pre-training is to derive the meta-defense policy specified in (3) for later online adaptation. Unlike finite Stackelberg Markov games that can be solved (approximately) using mixed-integer programming (Vorobeychik & Singh, 2021) or Q-learning (Sengupta & Kambhampati, 2020), our BSMG admits high-dimensional continuous state and action spaces, posing a more challenging computation issue. Hence, we resort to a two-timescale policy gradient (PG) algorithm, referred to as meta-Stackelberg learning (meta-SL) presented in Algorithm 1, to solve for the meta-SE in a similar vein to (Li et al., 2022b). In plain words, meta-SL first learns the attacker's best response at a fast scale (line 8-10), based on which updates the defender's meta policy at a slow scale at each iteration (line 13) using either debiased meta-

---

**Algorithm 1** Meta-Stackelberg Learning

1: **Input:** the distribution $Q(\xi)$, initial defense meta policy $\theta^0$, pre-trained attack policies $\{\phi_\xi^0\}_{\xi \in \Xi}$, step size parameters $\kappa_\mathcal{D}, \kappa_\mathcal{A}, \eta$, and iterations numbers $N_\mathcal{A}, N_\mathcal{D}$;
2: **Output:** $\theta^{N_\mathcal{D}}$;
3: **for** iteration $t = 0$ to $N_\mathcal{D} - 1$ **do**
4:     Sample a batch of attacks $\xi \in \hat{\Xi}$ from $Q$;
5:     **for** each sampled attack $\xi$ **do**
6:         Apply one-step adaptation $\theta_\xi^t \leftarrow \theta^t + \eta \hat{\nabla}_\theta J_\mathcal{D}(\theta^t, \phi_\xi^t, \xi)$;
7:         $\phi_\xi^t(0) \leftarrow \phi_\xi^t$;
8:         **for** iteration $k = 0, \ldots, N_\mathcal{A} - 1$ **do**
9:             $\phi_\xi^t(k+1) \leftarrow \phi_\xi^t(k) + \kappa_\mathcal{A} \hat{\nabla}_\phi J_\mathcal{A}(\theta_\xi^t, \phi_\xi^t(k), \xi)$;
10:         **end for**
11:         Estimate $\hat{\nabla} J_\mathcal{D}(\xi) := \hat{\nabla}_\theta J_\mathcal{D}(\theta, \phi_\xi^t(N_\mathcal{A}), \xi)|_{\theta = \theta_\xi^t}$;
12:     **end for**
13:     $\theta^{t+1} \leftarrow \texttt{Meta-Update}(\theta^t, \{\hat{\nabla} J_\mathcal{D}(\xi)\}_{\hat{\Xi}})$
14: **end for**

---

learning (Fallah et al., 2021a) or reptile (Nichol et al., 2018). The two-timescale meta-SL alleviates the nonstationarity caused by concurrent policy updates from both players (Yongacoglu et al., 2023). The exact formulation of the meta update rule and policy gradient estimation is deferred to Appendix B.

As shown in the algorithm, meta-SL requires interactions with attacks sampled from the attack domain to learn the meta-equilibrium. These interactions emulate the real FL process, thanks to the simulated environment (simulator) we construct in Section 4.1. However, these sampled attacks may not account for the true attack in the online execution, meaning that the meta policy is never exposed to such an attack, which poses an out-of-distribution (OOD) generalization issue (Fallah et al., 2021b) to the proposed meta-SG framework. Proposition 2.2 asserts that meta-SG is generalizable to the unseen attacks, given that the unseen is not distant from those seen. The formal statement is deferred to Appendix D, and the proof mainly targets those unseen non-adaptive attacks for simplicity.

**Proposition 2.2** (OOD Generalization)**.** *Consider sampled attack types $\xi_1, \ldots, \xi_m$ during the pre-training and the unseen attack type $\xi_{m+1}$ in the online stage. The generalization error is upper-bounded by the "discrepancy" between the unseen and the seen attacks $C(\xi_{m+1}, \{\xi_i\}_{i=1}^m)$.*

We finally conclude this section with a remark on the online adaptation practicality. During the online adaptation stage, the defender begins with the meta-policy learned from the pre-training stage to interact with the true FL environment, while collecting trajectories $\{s, \widetilde{r}, s'\}$. Here, the estimated reward $\widetilde{r}$ is calculated using the simulator (see Section 4.1). For a fixed period of FL epochs (e.g., 50 for MNIST and 100 for CIFAR-10), the defense policy will be updated using the collected trajectories. Ideally, the defender's adaptation time (including collecting samples and updating policy) should be significantly less than the whole FL training period so that the defense execution will not be delayed. In real-world FL training, the server typically waits for $1 \sim 10$ minutes before receiving responses from the clients (Bonawitz et al., 2019; Kairouz et al., 2021), which allows the defender to update the defense policy with enough episodes.

## 3 NON-ASYMPTOTIC COMPLEXITY OF META STACKELBERG LEARNING

This section presents the complexity results of meta-SL in Algorithm 1 using debiased meta-learning (Fallah et al., 2021a) as the updating rule, and detailed proofs can be found in Appendix D. Our analysis shows that the computation expense of the proposed meta-SL [$O(\epsilon^{-2})$ outer iterations; $O(\log \epsilon^{-1})$ inner iterations] does not differ much from that of meta-learning [$O(\epsilon^{-2})$, see (Fallah et al., 2021a)]. Weighing the marginal computation burden and the significant online adaptability showcased in Section 4, we recommend meta-SG in adversarial FL with intelligent adversaries.

We start our analysis with an alternative solution concept that is slightly weaker than Definition 2.1. To simplify our exposition, we let $\mathcal{L}_{\mathcal{D}}(\theta, \phi, \xi) := \mathbb{E}_{\tau \sim q} J_{\mathcal{D}}(\theta + \eta \hat{\nabla}_\theta J_{\mathcal{D}}(\tau), \phi, \xi)$ and $\mathcal{L}_{\mathcal{A}}(\theta, \phi, \xi) := \mathbb{E}_{\tau \sim q} J_{\mathcal{A}}(\theta + \hat{\nabla}_\theta J_{\mathcal{D}}(\tau), \phi, \xi)$, for a fixed type $\xi \in \Xi$. In the sequel, we will assume $\mathcal{L}_{\mathcal{D}}$ and $\mathcal{L}_{\mathcal{A}}$ to be continuously twice differentiable and Lipschitz-smooth with respect to both $\theta$ and $\phi$ as in (Li et al., 2022b), and the Lipschitz assumptions are deferred to Appendix D.

**Definition 3.1.** For a small $\varepsilon \in (0, 1)$, a set of parameters $(\theta^*, \{\phi_\xi^*\}_{\xi \in \Xi}) \in \Theta \times \Phi^{|\Xi|}$ is a $\varepsilon$-*meta First-Order Stackelbeg Equilibrium* ($\varepsilon$-meta-FOSE) of the meta-SG if it satisfies the following conditions for $\xi \in \Xi$, $\max_{\theta \in \Theta \cap B(\theta^*)} \langle \nabla_\theta \mathcal{L}_{\mathcal{D}}(\theta^*, \phi_\xi^*, \xi), \theta - \theta^* \rangle \leq \varepsilon$, $\max_{\phi \in \Phi \cap B(\phi_\xi^*)} \langle \nabla_\phi \mathcal{L}_{\mathcal{A}}(\theta^*, \phi_\xi^*, \xi), \phi - \phi_\xi^* \rangle \leq \varepsilon$, where $B(\theta^*) := \{\theta \in \Theta : \|\theta - \theta^*\| \leq 1\}$, and $B(\phi_\xi^*) := \{\theta \in \Theta : \|\phi - \phi_\xi^*\| \leq 1\}$. When $\varepsilon = 0$, the parameter set $(\theta^*, \{\phi_\xi^*\}_{\xi \in \Xi})$ is said to be the meta-FOSE.

Definition 3.1 contains the necessary equilibrium condition for Definition 2.1, which can be reduced to $\|\nabla_\theta \mathcal{L}_{\mathcal{D}}(\theta^*, \phi_\xi, \xi)\| \leq \varepsilon$ and $\|\nabla_\phi \mathcal{L}_{\mathcal{A}}(\theta^*, \phi_\xi, \xi)\| \leq \varepsilon$ in the unconstraint settings. Since we utilize stochastic gradient in practice, all inequalities mentioned above shall be considered in expectation. These conditions, along with the positive-semi-definiteness of the Hessians, construct the optimality conditions for a local solution for the meta-SE, which may not exist even in the zero-sum cases (Jin et al., 2019). Therefore, we limit our attention to the meta-FOSE whose existence is guaranteed by the following theorem.

**Theorem 3.2.** *Assuming that $\Theta$ and $\Phi$ are compact and convex, there exists at least one meta-FOSE.*

For the rest of this paper, we assume the attacker is unconstrained, i.e., $\Phi$ is a finite-dimensional Euclidean space to avoid discussing another projection operation in the attacker's gradient ascent.

**First-order Gradient Estimation.** To find a meta-FOSE for (3) is challenging since the lower-level problem involves a non-convex equilibrium constraint. To see this more clearly, consider differentiating the defender's value function: $\nabla_\theta V = \mathbb{E}_{\xi \sim Q}[\nabla_\theta \mathcal{L}_{\mathcal{D}}(\theta, \phi_\xi, \xi) + (\nabla_\theta \phi_\xi(\theta))^\top \nabla_\phi \mathcal{L}_{\mathcal{D}}(\theta, \phi_\xi, \xi)]$, where $\nabla_\theta \phi_\xi(\cdot)$ is locally characterized by the implicit function theorem, i.e., $\nabla_\theta \phi_\xi(\theta) = (-\nabla_\phi^2 \mathcal{L}_{\mathcal{A}}(\theta, \phi, \xi))^{-1} \nabla_{\phi\theta}^2 \mathcal{L}_{\mathcal{A}}(\theta, \phi, \xi)$. Therefore, the gradient estimation requires iteratively estimating the second-order information for the attacker (lower level) objective, which can be costly and

prohibitive in many scenarios (Song et al., 2019). Hence, We introduce the following assumption to bypass the technicality involved in calculating $\nabla_\theta \phi_\xi$, adapted from (Adler et al., 2009).

**Assumption 3.3** (Strict-Competitiveness). The BSMG is strictly competitive, i.e., there exist constants $c < 0$, $d$ such that $\forall \xi \in \Xi$, $s \in S$, $a_\mathcal{D}, a_\mathcal{A} \in A_\mathcal{D} \times A_\xi$, $r_\mathcal{D}(s, a_\mathcal{D}, a_\mathcal{A}) = cr_\mathcal{A}(s, a_\mathcal{D}, a_\mathcal{A}) + d$. One can treat the SC notion as a generalization of zero-sum games: if one joint action $(a_\mathcal{D}, a_\mathcal{A})$ leads to payoff increases for one player, it must decrease the other's payoff. In adversarial FL, the untargeted attack naturally makes the game zero-sum (hence, SC). The purpose of introducing Assumption 3.3 is to establish the Danskin-type result (Bernhard & Rapaport, 1995) for the Stackelberg game with nonconvex value functions (see Lemma 3.5), which spares us from the Hessian inversion.

In addition to the assumptions above, another regularity assumption we impose on the nonconvex value functions is adapted from the Polyak-Łojasiewicz (PL) condition (Karimi et al., 2016), which is customary in nonconvex analysis. Under Assumption 3.4, we are able to show the sufficiency of first-order estimation in Lemma 3.5, which subsequently leads to the main result in Theorem 3.6

**Assumption 3.4** (Stackelberg Polyak-Łojasiewicz condition). There exists a positive constant $\mu$ such that for any $(\theta, \phi) \in \Theta \times \Phi$ and $\xi \in \Xi$, the following inequalities hold: $\frac{1}{2\mu}\|\nabla_\phi \mathcal{L}_\mathcal{D}(\theta, \phi, \xi)\|^2 \geq \max_\phi \mathcal{L}_\mathcal{D}(\theta, \phi, \xi) - \mathcal{L}_\mathcal{D}(\theta, \phi, \xi)$, $\frac{1}{2\mu}\|\nabla_\phi \mathcal{L}_\mathcal{A}(\theta, \phi, \xi)\|^2 \geq \max_\phi \mathcal{L}_\mathcal{A}(\theta, \phi, \xi) - \mathcal{L}_\mathcal{A}(\theta, \phi, \xi)$.

**Lemma 3.5.** *Under Assumptions 3.4 and regularity conditions, there exists $\{\phi_\xi : \phi_\xi \in \arg\max_\phi \mathcal{L}_\mathcal{A}(\theta, \phi, \xi)\}_{\xi \in \Xi}$, such that $\nabla_\theta V(\theta) = \nabla_\theta \mathbb{E}_{\xi \sim Q, \tau \sim q} J_\mathcal{D}(\theta + \eta \hat{\nabla}_\theta J_\mathcal{D}(\tau), \phi_\xi, \xi)$. Moreover, there exists a constant $L > 0$ such that the defender value function $V(\theta)$ is L-Lipschitz-smooth.*

**Theorem 3.6.** *Under assumption 3.4 and regularity assumptions, for any given $\varepsilon \in (0, 1)$, let the learning rates $\kappa_\mathcal{A}$ and $\kappa_\mathcal{D}$ be properly chosen; let $N_\mathcal{A} \sim \mathcal{O}(\log \epsilon^{-1})$ and $N_b \sim \mathcal{O}(\epsilon^{-4})$ be properly chosen (Appendix D), then, Algorithm 1 finds a $\varepsilon$-meta-FOSE within $N_\mathcal{D} \sim \mathcal{O}(\varepsilon^{-2})$ iterations.*

## 4 EXPERIMENTS

### 4.1 EXPERIMENT SETTINGS

This section evaluates our meta-SG defense on MNIST (LeCun et al., 1998) and CIFAR-10 (Krizhevsky et al., 2009) datasets under several state-of-the-art attacks, including non-adaptive/adaptive untargeted model poison attacks (i.e., explicit boosting (EB) (Bhagoji et al., 2019), IPM (Xie et al., 2020), LMP (Fang et al., 2020), RL (Li et al., 2022a)), BFL (Bagdasaryan et al., 2020), DBA (Xie et al., 2019), PGD (Wang et al., 2020), BRL (Li et al., 2023)) and a mix of the two. We consider various strong defenses as baselines, including training-stage defenses such as Krum (Blanchard et al., 2017), Clipping Median (Yin et al., 2018; Sun et al., 2019; Li et al., 2022a), FLTrust (Cao et al., 2021), training stage CRFL (Xie et al., 2021) and post-training stage defenses such as Neuron Clipping (Wang et al., 2022) and Pruning (Wu et al., 2020). Compared with our meta-SG defense trained by adaptive attacks, we also consider a meta-learning defense presented in Section 2.3 (see Appendix B for more details), which is trained using a set of non-adaptive attacks. We use the following default parameters: number of devices $= 100$, number of malicious clients for untargeted model poisoning attack $= 20$, number of malicious clients for backdoor attack $= 10$, subsampling rate $= 10\%$, number of FL epochs $= 500$ (1000) for MNIST (CIFAR-10). The local data distributions across clients are assumed to be $i.i.d.$ in the default setting. We utilize the Twin Delayed DDPG (TD3) Fujimoto et al. (2018) algorithm to train both attacker's and defender's policies. Appendix E includes a detailed description of the experiment setup. Due to the space limit, additional experiment results and ablation studies are moved to Appendix F.

**Simulated Environment.** To simulate transitions and reward functions in BSMG, we first assume the defender always considers the worst-case scenario based on rough estimate about the number of malicious clients controlled by each attacker and non-$i.i.d.$ level of clients' local data distribution. For example, the defender will consider $40\%$ devices are malicious when the actual percentage varies from $10\%$ to $40\%$. Second, to simulate clients' behaviors (i.e., local training), the server needs a large amount of data, which is typically unavailable. We use inference attack (i.e., Inverting gradient (Geiping et al., 2020)) in (Li et al., 2022a) for only a few FL epochs (20 in our setting) to learn data from clients considering server can collect a group of gradients (10 in our setting) in each FL round. The server will then apply data augmentation (Shorten & Khoshgoftaar, 2019) to generate more data samples. We then use those data to train a conditional GAN model (Mirza & Osindero, 2014) for MNIST and a diffusion model (Sohl-Dickstein et al., 2015) for CIFAR-10 to generate as much data as necessary to simulate the local training in the simulated environment. In practice, the defender (i.e., server) does not know the backdoor attacker's triggers and/or targeted

labels. To simulate a backdoor attacker's behavior, we implement reverse engineering in Wang et al. (2019b) to reconstruct backdoor triggers that each target on one label and consider them as different types of attacks in the simulated environment. Since the defender does not know the poison ratio and target label of the attacker's poisoned dataset, we modify the defender's reward function as $r_{\mathcal{D}}^t = -\mathbb{E}[F''(\widehat{w}_g^{t+1})]$, $F''(w) := \lambda'F(w,U) - (1-\lambda')\min_{c \in C}[\frac{1}{|U'|}\sum_{j=1}^{|U'|}\ell(w,(\hat{x}_i^j,c))] \geq \lambda'F(w,U) - (1-\lambda')[\frac{1}{|U'|}\sum_{j=1}^{|U'|}\ell(w,(\hat{x}_i^j,c^*))]$, where $c^*$ is the truly targeted label, and $\lambda' \in [0,1]$ measures the tradeoff between the main task and the backdoor task. Here we assume all data in $U'$ are poisoned to approximate the true attack objective $\lambda F(w,U) + (1-\lambda)F(w,U')$ with another $\lambda$. Notice that even the same method is used to estimate the rewards in pre-training and online adaptation stages without knowing the exact attack, the server can collect each round's real FL model parameters as feedback to adapt the policy during online adaptation.

**Defense Action Compression.** Following the BSMG model, it is natural to use $w_g^t$ or $(w_g^t, \mathbf{I}^t)$ as the state, and $\{\widetilde{g}_k^t\}_{k=1}^{M_1+M_2}$ or $w_g^{t+1}$ as the action for the attacker and the defender, respectively, if the federated learning model is small. However, when we use federated learning to train a high-dimensional model (i.e., a large neural network), the original state/action space will lead to an extremely large search space that is prohibitive in terms of training time and memory space. To compress the defense action space against **untargeted model poisoning attacks**, we leverage the following robust aggregation based defenses: (1) coordinate-wise trimmed mean (Yin et al., 2018) with a trimming threshold $b = [0, \frac{1}{2})$ (dimension-wise); (2) clipping (Sun et al., 2019) with a norm bound $a$ (magnitude); and (3) FoolsGold (Fung et al., 2018) with a cosine similarity threshold $c$ (direction). These defenses are all training stage defenses. For **backdoor attacks**, we clip each model update with a norm bound of $a$ and then introduce Gaussian noise random noise to each coordinate with a variance $d$ as a training stage defense. Further, at the post-training stage, we consider Neuron Clipping with a clip range of $e$ or Pruning with a pruning mask rate of $f$. While the specific technique employed in each of these defenses could be substituted by other algorithms, the novelty of our approach lies in the utilization of RL to optimize them, as opposed to the conventional practice of using non-adaptive, handcrafted hyperparameters. That is, we consider $a_1^t := (b, a, c)$ as the action for untargeted defense and $a_2^t := (d, a, e/f)$ as the action for backdoor defense, which are obtained from the defense policy depending on the current state.

## 4.2 EXPERIMENT RESULTS

**Effectiveness against Non-adaptive/Adaptive attacks.** Our meta-SG defense is originally designed to defend mixed type attacks (Figure 1 (right)) and adaptive attacks (Figure 1 (left)) in the practical FL environment. However, with online adaptation, it can still reach the same level of state-of-the-art effectiveness against traditional single-type non-adaptive attacks as shown in Table 1 under untargeted model poisoning attacks (i.e., EB, IPM, LMP) and Table 2 under backdoor attacks (i.e., BFL, DBA, PGD). In the last rows of both tables, we demonstrate the superb performance of our meta-SG against RL-based attacks (i.e., RL, BRL). In fact, during online adaptation, the defender's problem against non-adaptive (resp. adaptive) attackers reduces to a single-player Markov Decision Process (resp. a two-player Markov Stackelberg Game). Once the defender has a simulated environment close to the real FL environment, the learned defense policy will be close to the optimal defense policy.

Table 1: Comparisons of average model accuracy (higher the better) after 500 FL rounds under untargeted model poisoning attacks and defenses on MNIST.

| Untarget | Krum | Clipping Median | FLtrust | Meta-SG (ours) |
|---|---|---|---|---|
| EB | $0.93(\pm0.02)$ | $0.94(\pm0.01)$ | $0.93(\pm0.03)$ | $0.95(\pm0.01)$ |
| IPM | $0.85(\pm0.05)$ | $0.87(\pm0.02)$ | $0.85(\pm0.04)$ | $0.85(\pm0.01)$ |
| LMP | $0.80(\pm0.02)$ | $0.76(\pm0.07)$ | $0.79(\pm0.02)$ | $0.81(\pm0.02)$ |
| RL | $0.12(\pm0.00)$ | $0.17(\pm0.04)$ | $0.45(\pm0.02)$ | $0.86(\pm0.02)$ |

**Adaptation to Uncertain/Unknown attacks.** To evaluate the efficiency of adaptation and examine the necessity of adapting from meta-SE policy, we introduce a meta-learning-based defense called meta-RL (see details in Appendix B), where the meta policy is trained over a set of non-adaptive attacks. As shown in Figure 1, our meta-SG can quickly adapt to both uncertain RL-based adaptive attack (attack action is time-varying during FL) and non-adaptive LMP attack, while meta-RL can only slowly adapt to or fail to adapt to the RL-based adaptive attacks on MNIST and CIFAT-10 respectively. Also, Figures 3 (a) and 3 (c) demonstrate the power of meta-SG against unknown LMP

Table 2: Comparisons of average backdoor accuracy (lower the better) after 500 FL rounds under backdoor attacks and defenses on CIFAR-10.

| Backdoor | Neuron Clipping | Pruning | CRFL | Meta-SG (ours) |
|---|---|---|---|---|
| BFL | 0.02($\pm$0.01) | 0.09($\pm$0.05) | 0.40($\pm$0.04) | 0.04($\pm$0.01) |
| DBA | 0.26($\pm$0.03) | 0.23($\pm$0.07) | 0.27($\pm$0.06) | 0.24($\pm$0.03) |
| PGD | 0.15($\pm$0.12) | 0.21($\pm$0.05) | 0.68($\pm$0.16) | 0.20($\pm$0.04) |
| BRL | 0.99($\pm$0.01) | 0.95($\pm$0.03) | 0.92($\pm$0.02) | 0.22($\pm$0.02) |

attack, even LMP is not directly used during its pre-training stage. Similar observations are given under IPM in Appendix F.

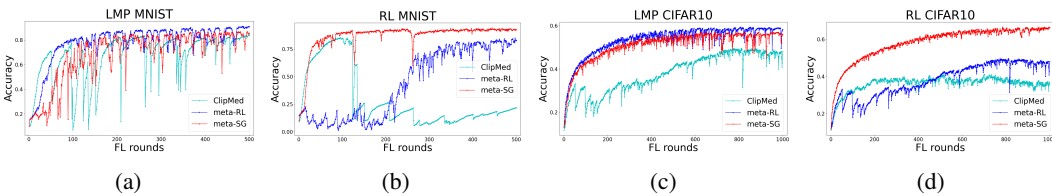

|  (a)  |  (b)  |  (c)  |  (d)  |

Figure 3: Comparisons of defenses against untargeted model poisoning attacks (i.e., LMP and RL) on MNIST and CIFAR-10. All parameters are set as default.

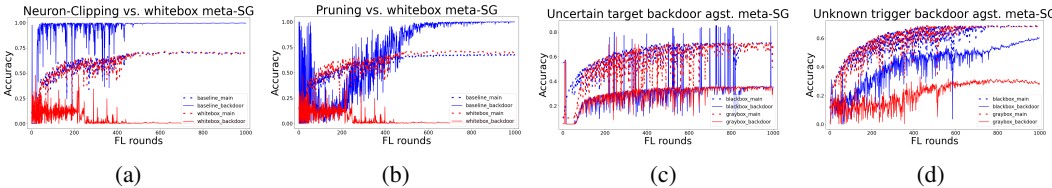

|  (a)  |  (b)  |  (c)  |  (d)  |

Figure 4: Comparisons of defenses (i.e., Neuron Clipping, Pruning, and meta-SG) under RL-based backdoor attack (BRL) on CIFAR-10. The BRLs are trained before epoch 0 against the associate defenses (i.e., Neuron Clipping, Pruning, and meta-policy of meta-SG). Other parameters are set as default.

**Defender's knowledge of backdoor attacks.** We consider two settings: 1) the server learned the backdoor trigger from reverse engineering (Wang et al., 2019b)) but is uncertain about the target label, and 2) the server knows the target label but not the backdoor trigger. In the former case, we generate triggers using reverse engineering targeting all 10 classes in CIFAR-10 in the simulated environment to train a defense policy in a **blackbox** setting, and reverse engineering targeting classes 0-4 in the simulated environment to train a defense policy in a **graybox** setting, respectively. We then apply a GAN-based model (Doan et al., 2021) targeting class 0 (airplane) to test the defense in each setting, with results shown in Figure 4(c). In the latter case where the defender does not know the true backdoor trigger used by the attacker, we implement the GAN-based models to randomly generate distributions of triggers (see Figure 6) targeting one known label (truck) to simulate a **blackbox** setting, as well as using reverse engineering (Wang et al., 2019b) targeting on one known label (truck) to simulate a **graybox** setting, and train a defense policy for each setting, and then apply a fixed global pattern (see Figure 7) in the real FL environment to test the defense (results shown in Figure 4(d)). In the **whitebox** setting, the server knows the backdoor trigger pattern (global) and the targeted label (truck), and corresponding results are in Figures 4(a) and 4(b). Post-training defenses alone, such as Neuron Clipping and Pruning, are susceptible to RL-based attacks once the defense mechanism is known. However, as depicted in Figure 4(a) and (b), we demonstrate that our whitebox meta-SG approach is capable of effectively eliminating the backdoor influence while preserving high main task accuracy simultaneously. Figure 4(c) illustrates that graybox meta-SG exhibits a more stable and robust mitigation of the backdoor attack compared to blackbox meta-SG. Furthermore, in Figure 4(d), graybox meta-SG demonstrates a significant reduction in the impact of the backdoor attack, achieving nearly a 70% mitigation, outperforming blackbox meta-SG.

## 5 CONCLUSION

We have proposed a meta-Stackelberg framework to tackle attacks of uncertain/unknown types in federated learning using data-driven adaptation, which is also relevant to a variety of security contexts with incomplete information regarding intelligent attackers. The proposed meta-equilibrium approach, computationally tractable and strategically adaptable, targets mixed and adaptive attacks under incomplete information. For discussions on broader impacts and limitations, see Appendix G.

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
