# A    RELATED WORKS

**Poisoning/backdoor attacks and defenses in FL.**    Various methods for compromising the integrity of a federated learning target model have been introduced, including targeted poisoning attacks which strive to misclassify a particular group of inputs, as explored in the studies by (Bhagoji et al., 2019; Baruch et al., 2019). Other techniques, such as those studied by (Fang et al., 2020; Xie et al., 2020; Shejwalkar & Houmansadr, 2021), focus on untargeted attacks with the aim of diminishing the overall model accuracy. The majority of existing strategies often utilize heuristics-based methods (e.g., (Xie et al., 2020)), or they focus on achieving a short-sighted goal ( (Fang et al., 2020; Shejwalkar & Houmansadr, 2021)). On the other hand, malicious participants can easily embed backdoors into the aggregated model while maintaining the model's performance on the main task with model replacement Bagdasaryan et al. (2020). To enhance the surreptitious nature of these poisoned updates, triggers can be distributed across multiple cooperative malicious devices, as discussed by Xie et al. (2019)(Xie et al., 2019), and edge-case backdoors can be employed, as demonstrated by Wang et al. (2020) (Wang et al., 2020). However, these methods can be sub-optimal, especially when there's a need to adopt a robust aggregation rule. Additionally, these traditional methods typically demand access to the local updates of benign agents or precise parameters of the global model for the upcoming round (Xie et al., 2020; Fang et al., 2020) in order to enact a significant attack. In contrast to these methods, RL-based approach Li et al. (2022a); Shen et al. (2021); Li et al. (2023) employs reinforcement learning for the attack, reducing the need for extensive global knowledge while focusing on a long-term attack goal.

Several defensive strategies have been suggested to counter model poisoning attacks, which broadly fall into two categories: those based on robust aggregation and those centered around detection. Robust-aggregation-based defenses encompass techniques such as dimension-wise filtering. These methods treat each dimension of local updates individually, as explored in studies by (Bernstein et al., 2018; Yin et al., 2018). Another strategy is client-wise filtering, the goal of which is to limit or entirely eliminate the influence of clients who might harbor malicious intent. This approach has been examined in the works of (Blanchard et al., 2017; Pillutla et al., 2022; Sun et al., 2019). Some defensive methods necessitate the server having access to a minimal amount of root data, as detailed in the study by Cao et al. (2021). Naive backdoor attacks are limited by even simple defenses like norm-bounding (Sun et al., 2019) and weak differential private (Geyer et al., 2017) defenses. Despite to the sophisticated design of state-of-the-art non-addaptive backdoor attacks against federated learning, post-training stage defenses (Wu et al., 2020; Nguyen et al., 2021; Rieger et al., 2022) can still effectively erase suspicious neurons/parameters in the backdoored model.

**Multi-agent meta learning.**    Meta-learning, and in particular meta-reinforcement-learning aim to create a generalizable policy that can fast adapt to new tasks by exploiting knowledge obtained from past tasks Duan et al. (2016); Finn et al. (2017). The early use cases of meta-learning have been primarily single-agent tasks, such as few-shot classification and single-agent RL Finn et al. (2017). A recent research thrust is to extend the meta-learning idea to multi-agent systems (MAS), which can be further categorized into two main directions: 1) distributed meta-learning in MAS Kayaalp et al. (2022); Zhang et al. (2022); 2) meta-learning for generalizable equilibrium-seeking Gupta et al. (2021); Harris et al. (2022); Zhao & Zhu (2022); Ge et al. (2023). The former focuses on a decentralized operation of meta-learning over networked computation units to reduce computation/storage expenses. The latter is better aligned with the original motivation of meta-learning, which considers how to solve a new game (or multi-agent decision-making) efficiently by reusing past experiences from similar occasions.

In stark contrast to the existing research efforts, our work leverages the adaptability of meta-learning to address information asymmetry in dynamic games of incomplete information, leading to a new equilibrium concept: meta-equilibrium (see Definition 2.1). What distinguishes our work from the aforementioned ones is that 1) every entity in our meta-SG is a self-interest player acting rationally without any coordination protocol; 2) meta-learning in our work is beyond a mere solver for computing long-established equilibria (e.g., Stackelberg equilibrium); it brings up a non-Bayesian approach to processing information in dynamic games (see Appendix C), which is computationally more tractable. This meta-equilibrium notion has been proven effective in combating information asymmetry in adversarial FL. Since asymmetric information is prevalent in security studies, our work can shed light on other related problems.

**First-order methods in bilevel optimization.**    The meta-SG problem in (3) amounts to a stochastic bilevel optimization. The meta-SL in Algorithm 1 admits a much simpler gradient estimation than

what one would often observe in the bilevel optimization literature (Chen et al., 2023; Kwon et al., 2023), where the gradient estimate for the upper-level problem involves a Hessian inverse (Chen et al., 2023) or some first-order correction terms (Kwon et al., 2023). The key intuition behind this simplicity lies in the strict competitiveness (see Assumption 3.3). Informally speaking, (3) is more akin to minimax programming (Nouiehed et al., 2019; Li et al., 2022b), even though it is a general-sum game. However, the data-driven meta-adaptation within the value function in (3) leads to a more involved gradient estimation. since the data induces extra randomness in addition to policy gradient estimates (Fallah et al., 2021a). Perhaps, the closest to our work is (Li et al., 2022b), where the authors investigate adversarial meta-RL and arrive at a similar Stackelberg formulation. However, Li et al. (2022b) consider a minimax relaxation to the original Stackelberg formulation, leading to simpler nonconvex programming. Our work is among the first endeavors to investigate fully first-order algorithms for solving general-sum Stackelberg games.

## B  ALGORITHMS

This section elaborates on meta-learning defense in equation 1 and meta-Stackelberg learning in equation 3. To begin with, we first review the policy gradient method Sutton et al. (2000) in RL and its Monte-Carlo estimation. To simplify our exposition, we fix the attacker's policy $\phi$, and then BSMG reduces to a single-agent MDP, where the optimal policy to be learned is the defender's $\theta$.

**Policy Gradient**    The idea of the policy gradient method is to apply gradient ascent to the value function $J_{\mathcal{D}}$. Following Sutton et al. (2000), we obtain $\nabla_\theta J_{\mathcal{D}} := \mathbb{E}_{\tau \sim q(\theta)}[g(\tau; \theta)]$, where $g(\tau; \theta) = \sum_{t=1}^{H} \nabla_\theta \log \pi(a_{\mathcal{D}}^t | s^t; \theta) R(\tau)$ and $R(\tau) = \sum_{t=1}^{H} \gamma^t r(s^t, a_{\mathcal{D}}^t)$. Note that for simplicity, we suppress the parameter $\phi, \xi$ in the trajectory distribution $q$, and instead view it as a function of $\theta$. In numerical implementations, the policy gradient $\nabla_\theta J_{\mathcal{D}}$ is replaced by its Monte-Carlo (MC) estimation using sample trajectory. Suppose a batch of trajectories $\{\tau_i\}_{i=1}^{N_b}$, and $N_b$ denotes the batch size, then the MC estimation is

$$\hat{\nabla}_\theta J_{\mathcal{D}}(\theta, \tau) := 1/N_b \sum_{\tau_i} g(\tau_i; \theta). \tag{B1}$$

The same deduction also holds for the attacker's problem when fixing the defense $\theta$.

**Meta-Learning FL Defense**    As discussed in Section 2.3, meta-learning-based defense (meta defense) mainly targets non-adaptive attack methods, where $\pi_{\mathcal{A}}(\cdot; \phi, \xi)$ is a pre-fixed attack strategy following some rulebook, such as IPM Xie et al. (2020) and LMP Fang et al. (2020). In this case, the BSMG reduces to single-agent MDP for the defender, where the transition kernel is determined by the attack method. Mathematically, the meta-defense problem is given by

$$\max_{\theta, \Psi} \mathbb{E}_{\xi \sim Q(\cdot)}[J_{\mathcal{D}}(\Psi(\theta, \tau), \phi, \xi)]. \tag{B2}$$

Since the attack type is hidden from the defender, the adaptation mapping $\Psi$ is usually defined in a data-driven manner. For example, $\Psi(\theta, \tau)$ can be defined as a one-step stochastic gradient update with learning rate $\eta$: $\Psi(\theta, \tau) = \theta + \eta \hat{\nabla} J_{\mathcal{D}}(\tau_\xi)$ Finn et al. (2017) or a recurrent neural network in Duan et al. (2016). This work mainly focuses on gradient adaptation for the purpose of deriving theoretical guarantees in Appendix D.

With the one-step gradient adaptation, the meta-defense problem in equation B2 can be simplified as

$$\max_\theta \mathbb{E}_{\xi \sim Q(\cdot)} \mathbb{E}_{\tau \sim q(\theta)}[J_{\mathcal{D}}(\theta + \eta \hat{\nabla}_\theta J_{\mathcal{D}}(\tau), \phi, \xi)]. \tag{B3}$$

Recall that the attacker's strategy is pre-determined, $\phi, \xi$ can be viewed as fixed parameters, and hence, the distribution $q$ is a function of $\theta$. To apply the policy gradient method to equation B3, one needs an unbiased estimation of the gradient of the objective function in equation B3. Consider the gradient computation using the chain rule:

$$\nabla_\theta \mathbb{E}_{\tau \sim q(\theta)}[J_{\mathcal{D}}(\theta + \eta \hat{\nabla}_\theta J_{\mathcal{D}}(\tau), \phi, \xi)]$$
$$= \mathbb{E}_{\tau \sim q(\theta)} \{ \underbrace{\nabla_\theta J_{\mathcal{D}}(\theta + \eta \hat{\nabla}_\theta J_{\mathcal{D}}(\tau), \phi, \xi)(I + \eta \hat{\nabla}_\theta^2 J_D(\tau))}_{①}$$
$$+ \underbrace{J_{\mathcal{D}}(\theta + \eta \hat{\nabla}_\theta J_{\mathcal{D}}(\tau)) \nabla_\theta \sum_{t=1}^{H} \pi(a^t | s^t; \theta)}_{②} \}. \tag{B4}$$

The first term results from differentiating the integrand $J_{\mathcal{D}}(\theta + \eta\hat{\nabla}_\theta J_{\mathcal{D}}(\tau), \phi, \xi)$ (the expectation is taken as integration), while the second term is due to the differentiation of $q(\theta)$. One can see from the first term that the above gradient involves a Hessian $\hat{\nabla}^2 J_{\mathcal{D}}$, and its sample estimate is given by the following. For more details on this Hessian estimation, we refer the reader to Fallah et al. (2021a).

$$\hat{\nabla}^2 J_{\mathcal{D}}(\tau) = \frac{1}{N_b} \sum_{i=1}^{N_b} [g(\tau_i; \theta)\nabla_\theta \log q(\tau_i; \theta)^\mathsf{T} + \nabla_\theta g(\tau_i; \theta)] \tag{B5}$$

Finally, to complete the sample estimate of $\nabla_\theta \mathbb{E}_{\tau \sim q(\theta)}[J_{\mathcal{D}}(\theta + \eta\hat{\nabla}_\theta J_{\mathcal{D}}(\tau), \phi, \xi)]$, one still needs to estimate $\nabla_\theta J_{\mathcal{D}}(\theta + \eta\hat{\nabla}_\theta J_{\mathcal{D}}(\tau), \phi, \xi)$ in the first term. To this end, we need to first collect a batch of sample trajectories $\tau'$ using the adapted policy $\theta' = \theta + \eta\hat{\nabla}_\theta J_D(\tau)$. Then, the policy gradient estimate of $\hat{\nabla}_\theta J_{\mathcal{D}}(\theta')$ proceeds as in equation B1. To sum up, constructing an unbiased estimate of equation B4 takes two rounds of sampling. The first round is under the meta policy $\theta$, which is used to estimate the Hessian equation B5 and to adapt the policy to $\theta'$. The second round aims to estimate the policy gradient $\nabla_\theta J_{\mathcal{D}}(\theta + \eta\hat{\nabla}_\theta J_{\mathcal{D}}(\tau), \phi, \xi)$ in the first term in equation B4.

In the experiment, we employ a first-order meta-learning algorithm called Reptile Nichol et al. (2018) to avoid the Hessian computation. The gist is to simply ignore the chain rule and update the policy using the gradient $\nabla_\theta J_{\mathcal{D}}(\theta', \phi, \xi)|_{\theta' = \theta + \eta\hat{\nabla}_\theta J_{\mathcal{D}}(\tau)}$. Naturally, without the Hessian term, the gradient in this update is biased, yet it still points to the ascent direction as argued in Nichol et al. (2018), leading to effective meta policy. The advantage of Reptile is more evident in multi-step gradient adaptation. Consider a $l$-step gradient adaptation, the chain rule computation inevitably involves multiple Hessian terms (each gradient step brings a Hessian term) as shown in (Fallah et al., 2021a, Theorem 2). In contrast, Reptile only requires first-order information, and the meta-learning algorithm ($l$-step adaptation) is given by Algorithm 2.

---

**Algorithm 2** Reptile Meta-Reinforcement Learning with $l$-step adaptation

---

1: **Input:** the type distribution $Q(\xi)$, step size parameters $\kappa, \eta$
2: **Output:** $\theta^T$
3: randomly initialize $\theta^0$
4: **for** iteration $t = 1$ to $T$ **do**
5:     Sample a batch $\hat{\Xi}$ of $K$ attack types from $Q(\xi)$;
6:     **for** each $\xi \in \hat{\Xi}$ **do**
7:         $\theta_\xi^t(0) \leftarrow \theta^t$
8:         **for** $k = 0$ to $l - 1$ **do**
9:             Sample a batch trajectories $\tau$ of the horizon length $H$ under $\theta_\xi^t(k)$;
10:             Evaluate $\hat{\nabla}_\theta J_{\mathcal{D}}(\theta_\xi^t(k), \tau)$ using MC in equation B1;
11:             $\theta_\xi^t(k+1) \leftarrow \theta_\xi^t(k) + \kappa\hat{\nabla}_\theta J_{\mathcal{D}}(\theta^t, \tau)$
12:         **end for**
13:     **end for**
14:     Update $\theta^{t+1} \leftarrow \theta^t + 1/K \sum_{\xi \in \hat{\Xi}} (\theta_\xi^t(l) - \theta^t)$;
15: **end for**

---

**Meta-Stackelberg Learning**   Recall that in meta-SE, the attacker's policy $\phi_\xi^*$ is not pre-fixed. Instead, it is the best response to the defender's adapted policy as shown in equation 3. To obtain this best response, one needs alternative training: fixing the defense policy, and applying gradient ascent to the attacker's problem until convergence. It should be noted that the proposed meta-SL utilizes the unbiased gradient estimation in equation B5, which paves the way for theoretical analysis in Appendix D. Yet, we turn to the Reptile to speed up pre-straining in the experiments. We present both algorithms in Algorithm 3, and only consider one-step adaptation for simplicity. The multi-step version is a straightforward extension of Algorithm 3.

## C   FURTHER JUSTIFICATION ON META EQUILIBRIUM

This section offers further justification for the meta-equilibrium, and we argue that meta-equilibrium provides a data-driven approach to address incomplete information in dynamic games. Note that information asymmetry is prevalent in the adversarial machine learning context, where the attacker

---

**Algorithm 3** (Reptile) Meta-Stackelberg Learning with one-step adaptation

---

1: **Input:** the type distribution $Q(\xi)$, initial defense meta policy $\theta^0$, pre-trained attack policies $\{\phi_\xi^0\}_{\xi \in \Xi}$, step size parameters $\kappa_\mathcal{D}$, $\kappa_\mathcal{A}$, $\eta$, and iterations numbers $N_\mathcal{A}$, $N_\mathcal{D}$;
2: **Output:** $\theta^{N_\mathcal{D}}$
3: **for** iteration $t = 0$ to $N_\mathcal{D} - 1$ **do**
4:     Sample a batch $\hat{\Xi}$ of $K$ attack types from $Q(\xi)$;
5:     **for** each $\xi \in \hat{\Xi}$ **do**
6:         Sample a batch of trajectories using $\phi^t$ and $\phi_\xi^t$;
7:         Evaluate $\hat{\nabla}_\theta J_D(\theta^t, \phi_\xi^t, \xi)$ using equation B1;
8:         Perform one-step adaptation $\theta_\xi^t \leftarrow \theta^t + \eta \hat{\nabla}_\theta J_D(\theta_\xi^t(k), \phi_\xi^t, \xi)$;
9:         $\phi_\xi^t(0) \leftarrow \phi_\xi^t$;
10:         **for** $k = 0, \ldots, N_\mathcal{A} - 1$ **do**
11:           Sample a batch of trajectories using $\theta_\xi^t$ and $\phi_\xi^t(k)$;
12:           $\phi_\xi^t(k+1) \leftarrow \phi_\xi^t(k) + \kappa_\mathcal{A} \hat{\nabla}_\phi J_\mathcal{A}(\theta_\xi^t, \phi_\xi^t(k), \xi)$;
13:         **end for**
14:         **if** Reptile **then**
15:           Sample a batch of trajectories using $\theta_\xi^t$ and $\phi_\xi^t(N_\mathcal{A})$;
16:           Evaluate $\hat{\nabla} J_D(\xi) := \hat{\nabla}_\theta J_\mathcal{D}(\theta, \phi_\xi^t(N_\mathcal{A}), \xi)|_{\theta=\theta_\xi^t}$ using equation B1;
17:         **else**
18:           Sample a batch of trajectories using $\theta^t$ and $\phi_\xi^t(N_\mathcal{A})$;
19:           Evaluate the Hessian using equation B5;
20:           Sample a batch of trajectories using $\theta_\xi^t$ and $\phi_\xi^t(N_\mathcal{A})$;
21:           Evaluate $\hat{\nabla} J_D(\xi) := \hat{\nabla}_\theta J_\mathcal{D}(\theta_\xi^t, \phi_\xi^t(N_\mathcal{A}), \xi)$ using equation B4;
22:         **end if**
23:         $\bar{\theta}_\xi^t \leftarrow \theta^t + \kappa_\mathcal{D} \hat{\nabla} J_D(\xi)$;
24:     **end for**
25:     $\theta^{t+1} \leftarrow \theta^t + 1/K \sum_{\xi \sim \hat{\Xi}} (\bar{\theta}_\xi^t - \theta_t), \phi_\xi^{t+1} \leftarrow \phi_\xi^t(N_\mathcal{A})$;
26: **end for**

---

enjoys an information advantage (e.g., the attacker's type). The proposed meta-equilibrium notion can shed light on these related problems beyond the adversarial FL context.

We begin with the insufficiency of Bayesian Stackelberg equilibrium equation 2 in handling information asymmetry, a customary solution concept in security studies Li et al. (2022d). One can see from equation 2 that such an equilibrium is of ex-ante type: the defender's strategy is determined before the game starts. It targets an "representative" attacker (an average of all types). As the game unfolds, new information regarding the attacker's private type is revealed (e.g., through the global model updates). However, this ex-ante strategy does not enable the defender to handle this emerging information as the game proceeds. Using game theory language, the defender fails to adapt its strategy in the interim stage.

To create interim adaptability in this dynamic game of incomplete information, one can consider introducing the belief system to capture the defender's learning process on the hidden type. Let $I^t$ be the defender's observations up to time $t$, i.e., $I^t := (s^k, a_\mathcal{D}^k)_{k=1}^t s^{t+1}$. Denote by $\mathcal{B}$ the belief generation operator $b^{t+1}(\xi) = \mathcal{B}[I^t]$. With the Bayesian equilibrium framework, the belief generation can be defined recursively as below

$$b^{t+1}(\xi) = \mathcal{B}[s^t, a_\mathcal{D}^t, b^t] := \frac{b^t(\xi)\pi_\mathcal{A}(a_\mathcal{A}^t | s^t; \xi)\mathcal{T}(s^{t+1} | s^t, a_\mathcal{A}^t, a_\mathcal{D}^t)}{\sum_{\xi'} b^t(\xi')\pi_\mathcal{A}(a_\mathcal{A}^t | s^t; \xi')\mathcal{T}(s^{t+1} | s^t, a_\mathcal{A}^t, a_\mathcal{D}^t)}. \tag{C1}$$

Since $b^t$ is the defender's belief on the hidden type at time $t$, its belief-dependent Markovian strategy is defined as $\pi_\mathcal{D}(s^t, b^t)$. Therefore, the interim equilibrium, also called Perfect Bayesian Equilibrium

(PBE) Fudenberg & Tirole (1991) is given by a tuple $(\pi_{\mathcal{D}}^*, \pi_{\mathcal{A}}^*, \{b^t\}_{t=1}^H)$ satisfying

$$\pi_{\mathcal{D}}^* = \arg\max \mathbb{E}_{\xi \sim Q} \mathbb{E}_{\pi_{\mathcal{D}}, \pi_{\mathcal{A}}^*} [\sum_{t=1}^H r_{\mathcal{D}}(s^t, a_{\mathcal{D}}^t, a_{\mathcal{A}}^t) b^t(\xi)]$$

$$\pi_{\mathcal{A}}^* = \arg\max \mathbb{E}_{\pi_{\mathcal{D}}, \pi_{\mathcal{A}}} [\sum_{t=1}^H r_{\mathcal{A}}(s^t, a_{\mathcal{D}}^t, a_{\mathcal{A}}^t)], \forall \xi, \tag{PBE}$$

$\{b^k\}_{k=1}^H$ satisfies $(C1)$ for realized actions and states.

In contrast with (2), this perfect Bayesian equilibrium notion (PBE) enables the defender to make good use of the information revealed by the attacker, and subsequently adjust its actions according to the revealed information through the belief generation. From a game-theoretic viewpoint, both (PBE) and (3) create strategic online adaptation: the defender can infer and adapt to the attacker's private type through the revealed information since different types aim at different objectives, hence, leading to different actions. Compared with PBE, the proposed meta-equilibrium notion is better suited for large-scale complex systems where players' decision variables can be high-dimensional and continuous, as argued in the ensuing paragraph.

To achieve the strategic adaptation, PBE relies on the Bayesian-posterior belief updates, which soon become intractable as the denominator in equation C1 involves integration over high-dimensional space and discretization inevitably leads to the curse of dimensionality. Despite the limited practicality, PBE is inherently difficult to solve, even in finite-dimensional cases. It is shown in Bhaskar et al. (2016) that the equilibrium computation in games with incomplete information is NP-hard, and how to solve for PBE in dynamic games remains an open problem. Even though there have been encouraging attempts at solving PBE in two-stage games Li & Zhu (2023), it is still challenging to address PBE computation in generic Markov games.

## D THEORETICAL RESULTS

### D.1 EXISTENCE OF META-SG

**Theorem D.1** (Theorem 3.2). *Under the conditions that $\Theta$ and $\Phi$ are compact and convex, the meta-SG admits at least one meta-FOSE.*

*Proof.* Clearly, $\Theta \times \Phi^{|\Xi|}$ is compact and convex, let $\phi \in \Phi^{|\Xi|}, \phi_\xi \in \Phi$ be the (type-aggregated) attacker's strategy, since the consider twice continuously differentiable utility functions $\ell_{\mathcal{D}}(\theta, \phi) := \mathbb{E}_{\xi \sim Q} \mathcal{L}_{\mathcal{D}}(\theta, \phi_\xi, \xi)$ and $\ell_\xi(\theta, \phi) := \mathcal{L}_{\mathcal{A}}(\theta, \phi_\xi, \xi)$ for all $\xi \in \Xi$. Then, there exists a constant $\gamma_c > 0$, such that the auxiliary utility functions:

$$\tilde{\ell}_{\mathcal{D}}(\theta; (\theta', \phi')) \equiv \ell_{\mathcal{D}}(\theta, \phi) - \frac{\gamma_c}{2} \|\theta - \theta'\|^2$$

$$\tilde{\ell}_\xi(\phi_\xi; (\theta', \phi') \equiv \ell_\xi(\theta', (\phi_\xi, \phi'_{-\xi})) - \frac{\gamma_c}{2} \|\phi_\xi - \phi'_\xi\|^2 \quad \forall \xi \in \Xi \tag{D2}$$

are $\gamma_c$-strongly concave in spaces $\theta \in \Theta, \phi_\xi \in \Phi$ for all $\xi \in \Xi$, respectively for any fixed $(\theta', \phi') \in \Theta \times \Phi^{|\Xi|}$.

Define the self-map $h : \Theta \times \Phi^{|\Xi|} \to \Theta \times \Phi^{|\Xi|}$ with $h(\theta', \phi') \equiv (\bar{\theta}(\theta', \phi'), \bar{\phi}(\theta', \phi'))$, where

$$\bar{\theta}(\theta', \phi') = \arg\max_{\theta \in \Theta} \tilde{\ell}_{\mathcal{D}}(\theta, \phi'), \qquad \bar{\phi}_\xi(\theta', \phi') = \arg\max_{\phi_\xi \in \Phi} \tilde{\ell}_\xi(\theta', (\phi_\xi, \phi'_{-\xi})).$$

Due to compactness, $h$ is well-defined. By strong concavity of $\tilde{\ell}_{\mathcal{D}}(\cdot; (\theta', \phi'))$ and $\tilde{\ell}_\xi(\cdot; (\theta', \phi'))$, it follows that $\bar{\theta}, \bar{\phi}$ are continuous self-mapping from $\Theta \times \Phi^{|\Xi|}$ to itself. By Brouwer's fixed point theorem, there exists at least one $(\theta^*, \phi^*) \in \Theta \times \Phi^{|\Xi|}$ such that $h(\theta^*, \phi^*) = (\theta^*, \phi^*)$. Then, one can verify that $(\theta^*, \phi^*)$ is a meta-FOSE of the meta-SG with utility function $\ell_{\mathcal{D}}$ and $\ell_\xi, \xi \in \Xi$, in view of the following inequality

$$\langle \nabla_\theta \tilde{\ell}_{\mathcal{D}}(\theta^*; (\theta^*, \phi^*)), \theta - \theta^* \rangle = \langle \nabla_\theta \ell_{\mathcal{D}}(\theta^*, \phi^*), \theta - \theta^* \rangle$$

$$\langle \nabla_{\phi_\xi} \tilde{\ell}_\xi(\theta^*; (\theta^*, \phi^*)), \phi_\xi - \phi_\xi^* \rangle = \langle \nabla_{\phi_\xi} \ell_\xi(\theta^*, \phi^*), \phi_\xi - \phi_\xi^* \rangle,$$

therefore, the equilibrium conditions for meta-SG with utility functions $\tilde{\ell}_{\mathcal{D}}$ and $\{\tilde{\ell}_\xi\}_{\xi \in \Xi}$ are the same as with utility functions $\ell_{\mathcal{D}}$ and $\{\ell_\xi\}_{\xi \in \Xi}$, hence the claim follows. $\square$

### D.2 PROOFS: NON-ASYMPTOTIC ANALYSIS

In the sequel, we make the following smoothness assumptions for every attack type $\xi \in \Xi$. In addition, we assume, for analytical simplicity, that all types of attackers are unconstrained, i.e., $\Phi$ is the Euclidean space with proper finite dimension.

**Assumption D.2** (($\xi$-wise) Lipschitz smoothness). *The functions $\mathcal{L}_\mathcal{D}$ and $\mathcal{L}_\mathcal{A}$ are continuously diffrentiable in both $\theta$ and $\phi$. Furthermore, there exists constants $L_{11}, L_{12}, L_{21},$ and $L_{22}$ such that for all $\theta, \theta_1, \theta_2 \in \Theta$ and $\phi, \phi_1, \phi_2 \in \Phi$, we have, for any $\xi \in \Xi$,*

$$\|\nabla_\theta \mathcal{L}_\mathcal{D}(\theta_1, \phi, \xi) - \nabla_\theta \mathcal{L}_\mathcal{D}(\theta_2, \phi, \xi)\| \leq L_{11} \|\theta_1 - \theta_2\| \tag{D3}$$

$$\|\nabla_\phi \mathcal{L}_\mathcal{D}(\theta, \phi_1, \xi) - \nabla_\phi \mathcal{L}_\mathcal{D}(\theta, \phi_2, \xi)\| \leq L_{22} \|\phi_1 - \phi_2\| \tag{D4}$$

$$\|\nabla_\theta \mathcal{L}_\mathcal{D}(\theta, \phi_1, \xi) - \nabla_\theta \mathcal{L}_\mathcal{D}(\theta, \phi_2, \xi)\| \leq L_{12} \|\phi_1 - \phi_2\| \tag{D5}$$

$$\|\nabla_\phi \mathcal{L}_\mathcal{D}(\theta_1, \phi, \xi) - \nabla_\phi \mathcal{L}_\mathcal{D}(\theta_2, \phi, \xi)\| \leq L_{12} \|\theta_1 - \theta_2\| \tag{D6}$$

$$\|\nabla_\phi \mathcal{L}_\mathcal{A}(\theta, \phi_1, \xi) - \nabla_\phi \mathcal{L}_\mathcal{A}(\theta, \phi_2, \xi)\| \leq L_{21} \|\phi_1 - \phi_2\|. \tag{D7}$$

**Lemma D.3** (Implicit Function Theorem (IFT) for Meta-SG). *Suppose for $(\bar{\theta}, \bar{\phi}) \in \Theta \times \Phi^{|\Xi|}$, $\xi \in \Xi$ we have $\nabla_\phi \mathcal{L}_\mathcal{A}(\bar{\theta}, \bar{\phi}, \xi) = 0$ the Hessian $\nabla_\phi^2 \mathcal{L}_\mathcal{A}(\bar{\theta}, \bar{\phi}, \xi)$ is non-singular. Then, there exists a neighborhood $B_\varepsilon(\bar{\theta}), \varepsilon > 0$ centered around $\bar{\theta}$ and a $C^1$-function $\phi(\cdot) : B_\varepsilon(\bar{\theta}) \to \Phi^{|\Xi|}$ such that near $(\bar{\theta}, \bar{\phi})$ the solution set $\{(\theta, \phi) \in \Theta \times \Phi^{|\Xi|} : \nabla_\phi \mathcal{L}_\mathcal{A}(\theta, \phi, \xi) = 0\}$ is a $C^1$-manifold locally near $(\bar{\theta}, \bar{\phi})$. The gradient $\nabla_\theta \phi(\theta)$ is given by $-(\nabla_\phi^2 \mathcal{L}_\mathcal{A}(\theta, \phi, \xi))^{-1} \nabla_{\phi\theta}^2 \mathcal{L}_\mathcal{A}(\theta, \phi, \xi)$.*

**Lemma D.4.** *Under assumptions D.2, 3.4, there exists $\{\phi_\xi : \phi_\xi \in \arg\max_\phi \mathcal{L}_\mathcal{A}(\theta, \phi, \xi)\}_{\xi \in \Xi}$, such that*

$$\nabla_\theta V(\theta) = \nabla_\theta \mathbb{E}_{\xi \sim Q, \tau \sim q} J_\mathcal{D}(\theta + \eta \hat{\nabla}_\theta J_\mathcal{D}(\tau), \phi_\xi, \xi).$$

*Moreover, the function $V(\theta)$ is $L$-Lipschitz-smooth, where $L = L_{11} + \frac{L_{12} L_{21}}{\mu}$*

$$\|\nabla_\theta V(\theta_1) - \nabla_\theta V(\theta_2)\| \leq L \|\theta_1 - \theta_2\|.$$

*Proof of Lemma D.4.* First, we show that for any $\theta_1, \theta_2 \in \Theta, \xi \in \Xi,$ and $\phi_1 \in \arg\max_\phi \mathcal{L}_\mathcal{A}(\theta_1, \phi, \xi)$, there exists $\phi_2 \in \arg\max_\phi \mathcal{L}_\mathcal{A}(\theta_2, \phi, \xi)$ such that $\|\phi_1 - \phi_2\| \leq \frac{L_{12}}{\mu} \|\theta_1 - \theta_2\|$. Indeed, based on smoothness assumption equation D7 and equation D6,

$$\|\nabla_\phi \mathcal{L}_\mathcal{A}(\theta_1, \phi_1, \xi) - \nabla_\phi \mathcal{L}_\mathcal{A}(\theta_2, \phi_1, \xi)\| \leq L_{21} \|\theta_1 - \theta_2\|,$$

$$\|\nabla_\phi \mathcal{L}_\mathcal{D}(\theta_1, \phi_1, \xi) - \nabla_\phi \mathcal{L}_\mathcal{D}(\theta_2, \phi_1, \xi)\| \leq L_{12} \|\theta_1 - \theta_2\|.$$

Since $\phi_2 \in \arg\max_\phi \mathcal{L}_\mathcal{A}(\theta_2, \phi, \xi), \nabla_\phi \mathcal{L}_\mathcal{A}(\theta_2, \phi_2, \xi) = 0$. Apply PL condition to $\nabla_\phi \mathcal{L}_\mathcal{A}(\theta, \phi_2, \xi)$,

$$\max_\phi \mathcal{L}_\mathcal{A}(\theta_1, \phi, \xi) - \mathcal{L}_\mathcal{A}(\theta_1, \phi_2, \xi) \leq \frac{1}{2\mu} \|\nabla_\phi \mathcal{L}_\mathcal{A}(\theta_1, \phi_2, \xi)\|^2$$

$$= \frac{1}{2\mu} \|\nabla_\phi \mathcal{L}_\mathcal{A}(\theta_1, \phi_2, \xi) - \nabla_\phi \mathcal{L}_\mathcal{A}(\theta_2, \phi_2, \xi)\|^2$$

$$\leq \frac{L_{21}^2}{2\mu} \|\theta_1 - \theta_2\|^2 \qquad \text{by equation D7.}$$

Since PL condition implies quadratic growth, we also have

$$\mathcal{L}_\mathcal{A}(\theta_1, \phi_1, \xi) - \mathcal{L}_\mathcal{A}(\theta_1, \phi_2, \xi) \geq \frac{\mu}{2} \|\phi_1 - \phi_2\|^2.$$

Combining the two inequalities above we obtain the Lipschitz stability for $\phi_\xi^*(\cdot)$, i.e.,

$$\|\phi_1 - \phi_2\| \leq \frac{L_{21}}{\mu} \|\theta_1 - \theta_2\|.$$

Second, show that $\nabla_\theta V(\theta)$ can be directly evaluated at $\{\phi_\xi^*\}_{\xi \in \Xi}$. Inspired by Danskin's theorem, we first made the following argument, consider the definition of directional derivative. Let $\ell(\theta, \phi) := \nabla_\theta \mathbb{E}_{\xi, \tau} J_\mathcal{D}(\theta + \eta \hat{\nabla} J_\mathcal{D}(\tau), \xi)$. For a constant $\tau$ and an arbitrary direction $d$,

$$\ell(\theta + \tau d, \phi^*(\theta + \tau d)) - \ell(\theta, \phi^*(\theta)))$$

$$= \ell(\theta + \tau d, \phi^*(\theta + \tau d)) - \ell(\theta + \tau d, \phi^*(\theta)) + \ell(\theta + \tau d, \phi^*(\theta)) - \ell(\theta, \phi^*(\theta))$$

$$= \nabla_\phi \ell(\theta + \tau d, \phi^*(\theta))^\top \underbrace{[\phi^*(\theta + \tau d) - \phi^*(\theta))]}_{\Delta\phi} + o(\Delta\phi^2)$$

$$+ \tau \nabla_\theta \ell(\theta, \phi^*(\theta))^T d + o(d^2).$$

Hence, a sufficient condition for the first equation is $\nabla_\phi \ell(\theta + \tau d, \phi^*(\theta)) = 0$, meaning that $\ell_D(\theta, \phi)$ and $\mathcal{L}_\mathcal{A}(\theta, \phi, \xi)$ share the first-order stationarity at every $\phi$ when fixing $\theta$. Indeed, by Lemma D.3, we have, the gradient is locally determined by

$$
\begin{aligned}
\nabla_\theta V &= \mathbb{E}_{\xi \sim Q}[\nabla_\theta \mathcal{L}_\mathcal{D}(\theta, \phi_\xi, \xi) + (\nabla_\theta \phi_\xi(\theta))^\top \nabla_\phi \mathcal{L}_\mathcal{D}(\theta, \phi_\xi, \xi)] \\
&= \mathbb{E}_{\xi \sim Q}\left[\nabla_\theta \mathcal{L}_\mathcal{D}(\theta, \phi_\xi, \xi) - [(\nabla_\phi^2 \mathcal{L}_\mathcal{A}(\theta, \phi, \xi))^{-1} \nabla_{\phi\theta}^2 \mathcal{L}_\mathcal{A}(\theta, \phi, \xi)]^\top \nabla_\phi \mathcal{L}_\mathcal{D}(\theta, \phi_\xi, \xi)\right].
\end{aligned}
$$

Given a trajectory $\tau := (s^1, a_\mathcal{D}^t, a_\mathcal{A}^t, \ldots, a_\mathcal{D}^H, a_\mathcal{A}^H, s^{H+1})$, let $R_\mathcal{D}(\tau, \xi) := \sum_{t=1}^H \gamma^{t-1} r_\mathcal{D}(s_t, a_t, \xi)$ and $R_\mathcal{D}(\tau, \xi) := \sum_{t=1}^H \gamma^{t-1} r_\mathcal{D}(s_t, a_t, \xi)$. Denote by $\mu(\tau; \theta, \phi)$ the trajectory distribution, that the log probability of $\mu$ is given by

$$
\log \mu(\tau; \theta, \phi) = \sum_{t=1}^H (\log \pi_\mathcal{D}(a_\mathcal{D}^t | s^t; \theta + \eta \hat{\nabla}_\theta J_\mathcal{D}(\tau)) + \log \pi_\mathcal{A}(a_\mathcal{A}^t | s^t; \phi) + \log P(s^{t+1} | a_\mathcal{D}^t, a_\mathcal{A}^t, s^t)
$$

According to the policy gradient theorem, we have

$$
\nabla_\phi \mathcal{L}_\mathcal{D}(\theta, \phi, \xi) = \mathbb{E}_\mu[R_\mathcal{D}(\tau, \xi) \sum_{t=1}^H \nabla_\phi \log(\pi_\mathcal{A}(a_\mathcal{A}^t | s^t; \phi))],
$$

$$
\nabla_\phi \mathcal{L}_\mathcal{A}(\theta, \phi, \xi) = \mathbb{E}_\mu[R_\mathcal{A}(\tau, \xi) \sum_{t=1}^H \nabla_\phi \log(\pi_\mathcal{A}(a_\mathcal{A}^t | s^t; \phi))].
$$

By SC Assumption 3.3, when $\nabla_\phi \mathcal{L}_\mathcal{A}(\theta, \phi, \xi) = 0$, there exists $c < 0$, $d$, such that $\nabla_\phi \mathcal{L}_\mathcal{D}(\theta, \phi, \xi) = \mathbb{E}_\mu[cR_\mathcal{A}(\tau, \xi) \sum_{t=1}^H \nabla_\phi \log(\pi_\mathcal{A}(a_\mathcal{A}^t | s^t; \phi))] + \mathbb{E}_\mu[\sum_{t=1}^H \gamma^{t-1} d \sum_{t=1}^H \nabla_\phi \log(\pi_\mathcal{A}(a_\mathcal{A}^t | s^t; \phi))] = 0$. Hence $\nabla_\theta V = \mathbb{E}_{\xi \sim Q}[\nabla_\theta \mathcal{L}_\mathcal{D}(\theta, \phi_\xi, \xi)]$.

Third, $V(\theta)$ is also Lipschitz smooth. As we notice that, $\ell_\mathcal{D}$ is Lipschitz smooth since $\mathbb{E}_{\xi \sim Q}$ is a linear operator, we have,

$$
\begin{aligned}
&\|\nabla_\theta V(\theta_1) - \nabla_\theta V(\theta_2)\| \\
&\leq \|\nabla_\theta \mathbb{E}_{\xi \sim Q} \mathcal{L}_\mathcal{D}(\theta_1, \phi_1, \xi) - \nabla_\theta \mathbb{E}_{\xi \sim Q} \mathcal{L}_\mathcal{D}(\theta_2, \phi_2, \xi)\| \\
&= \|\nabla_\theta \ell_\mathcal{D}(\theta_1, \phi_1) - \nabla_\theta \ell_\mathcal{D}(\theta_2, \phi_1) + \nabla_\theta \ell_\mathcal{D}(\theta_2, \phi_1) - \nabla_\theta \ell_\mathcal{D}(\theta_2, \phi_2)\| \\
&\leq \|\nabla_\theta \ell_\mathcal{D}(\theta_1, \phi_1) - \nabla_\theta \ell_\mathcal{D}(\theta_2, \phi_1)\| + \|\nabla_\theta \ell_\mathcal{D}(\theta_2, \phi_1) - \nabla_\theta \ell_\mathcal{D}(\theta_2, \phi_2)\| \\
&\leq L_{11}\|\theta_1 - \theta_2\| + L_{12}\|\phi_1 - \phi_2\| \\
&\leq (L_{11} + \frac{L_{12}L_{21}}{\mu})\|\theta_1 - \theta_2\|,
\end{aligned}
$$

which implies the Lipschitz constant $L = L_{11} + \frac{L_{12}L_{21}}{\mu}$. □

It is impossible to present the convergence theory without the assistance of some standard assumptions in batch reinforcement learning, of which the justification can be found in (Fallah et al., 2021a). We also require some additional information about the parameter space and function structure. These assumptions are all stated in Assumption D.5.

**Assumption D.5.**

(a) The following policy gradients are bounded, $\|\nabla_\phi \mathcal{L}_\mathcal{D}(\theta, \phi, \xi)\| \leq G^2$, $\|\mathcal{L}_\mathcal{A}(\theta, \phi, \xi)\| \leq G^2$ for all $\theta, \phi \in \Theta \times \Phi$ and $\xi \in \Xi$.

(b) The policy gradient estimations are unbiased.

(c) The variances for the stochastic gradients are bounded, i.e., for all $theta_\xi^t, \phi_\xi^t, \xi$,

$$
\mathbb{E}[\|\hat{\nabla}_\phi J(\theta_\xi^t, \phi_\xi^t, \xi) - \nabla_\phi J(\theta_\xi^t, \phi_\xi^t, \xi)\|^2] \leq \frac{\sigma^2}{N_b}.
$$

(d) The parameter space $\Theta$ has diameter $D_\Theta := \sup_{\theta_1, \theta_2 \in \Theta} \|\theta_1 - \theta_2\|$; the initialization $\theta^0$ admits at most $D_V$ function gap, i.e., $D_V := \max_{\theta \in \Theta} V(\theta) - V(\theta^0)$.

(e) It holds that the parameters satisfy $0 < \mu < -cL_{22}$.

Equipped with Assumption D.5 we are able to unfold our main result Theorem 3.6, before which we show in Lemma D.6 that $\phi_\xi^*$ can be efficiently approximated by the inner loop in the sense that $\nabla_\theta \mathbb{E}_{\xi \sim Q} \mathcal{L}_\mathcal{D}(\theta^t, \phi_\xi^t(N_\mathcal{A}), \xi) \approx \nabla_\theta V(\theta^t)$, where $\phi_\xi^t(N_\mathcal{A})$ is the last iterate output of the attacker policy.

**Lemma D.6.** *Under Assumption D.5, 3.4, 3.3, and D.2, let $\rho := 1 + \frac{\mu}{cL_{22}} \in (0,1)$, $\bar{L} = \max\{L_{11}, L_{12}, L_{22}, L_{21}, V_\infty\}$ where $V_\infty := \max\{\max \|\nabla V(\theta)\|, 1\}$. For all $\varepsilon > 0$, if the attacker learning iteration $N_\mathcal{A}$ and batch size $N_b$ are large enough such that*

$$N_\mathcal{A} \geq \frac{1}{\log \rho^{-1}} \log \frac{32 D_V^2 (2V_\infty + LD_\Theta)^4 \bar{L} |c| G^2}{L^2 \mu^2 \varepsilon^4}$$

$$N_b \geq \frac{32 \mu L_{21}^2 D_V^2 (2V_\infty + LD_\Theta)^4}{|c| L_{22}^2 \sigma^2 \bar{L} L \varepsilon^4},$$

*then, for $z_t := \nabla_\theta \mathbb{E}_{\xi \sim Q} \mathcal{L}_\mathcal{D}(\theta^t, \phi_\xi^t(N_\mathcal{A}), \xi) - \nabla_\theta V(\theta^t)$,*

$$\mathbb{E}[\|z_t\|] \leq \frac{L\varepsilon^2}{4 D_V (2V_\infty + LD_\Theta)^2},$$

*and*

$$\mathbb{E}[\|\nabla_\phi \mathcal{L}_\mathcal{A}(\theta^t, \phi_\xi^t(N), \xi)\|] \leq \varepsilon.$$

*Proof of Lemma D.6.* Fixing a $\xi \in \Xi$, due to Lipschitz smoothness,

$$\mathcal{L}_\mathcal{D}(\theta^t, \phi_\xi^t(N), \xi) - \mathcal{L}_\mathcal{D}(\theta^t, \phi_\xi^t(N-1), \xi)$$
$$\leq \langle \nabla_\phi \mathcal{L}_\mathcal{D}(\theta^t, \phi_\xi^t(N-1), \xi), \phi_\xi^t(N) - \phi_\xi^t(N-1) \rangle + \frac{L_{22}}{2} \|\phi_\xi^t(N) - \phi_\xi^t(N-1)\|^2.$$

The inner loop updating rule ensures that when $\kappa_\mathcal{A} = \frac{1}{L_{21}}$, $\phi_\xi^t(N) - \phi_\xi^t(N-1) = \frac{1}{L_{21}} \hat{\nabla}_\phi J_\mathcal{A}(\theta_\xi^t, \phi_\xi^t(N-1), \xi)$. Plugging it into the inequality, we arrive at

$$\mathcal{L}_\mathcal{D}(\theta^t, \phi_\xi^t(N), \xi) - \mathcal{L}_\mathcal{D}(\theta^t, \phi_\xi^t(N-1), \xi)$$
$$\leq \frac{1}{L_{21}} \langle \nabla_\phi \mathcal{L}_\mathcal{D}(\theta^t, \phi_\xi^t(N-1), \xi), \hat{\nabla}_\phi J_\mathcal{A}(\theta_\xi^t, \phi_\xi^t(N-1), \xi) \rangle + \frac{L_{22}}{2L_{21}^2} \|\hat{\nabla}_\phi J_\mathcal{A}(\theta_\xi^t, \phi_\xi^t(N-1), \xi)\|^2.$$

Therefore, we let $(\mathcal{F}_n^t)_{0 \leq n \leq N}$ be the filtration generated by $\sigma(\{\phi_\xi^t(\tau)\}_{\xi \in \Xi} | \tau \leq n)$ and take conditional expectations on $\mathcal{F}_n^t$:

$$\mathbb{E}[V(\theta^t) - \ell_\mathcal{D}(\theta^t, \phi^t(N)) | \mathcal{F}_{N-1}^t] \leq V(\theta^t) - \ell_\mathcal{D}(\theta^t, \phi^t(N-1))$$
$$\mathbb{E}_\xi \left[ \frac{1}{L_{21}} \langle \nabla_\phi \mathcal{L}_\mathcal{D}, \nabla_\phi J_\mathcal{A}(\theta_\xi^t, \phi_\xi^t(N-1), \xi) \rangle + \frac{L_{22}}{2L_{21}^2} \|\hat{\nabla}_\phi J_\mathcal{A}(\theta_\xi^t, \phi_\xi^t(N-1), \xi)\|^2 \right].$$

By variance-bias decomposition, and Assumption D.5 (b) and (c),

$$\mathbb{E}[\|\hat{\nabla}_\phi J_\mathcal{A}(\theta_\xi^t, \phi_\xi^t(N-1), \xi)\|^2 | \mathcal{F}_{N-1}^t]$$
$$= \mathbb{E}[\|\hat{\nabla}_\phi J_\mathcal{A}(\theta_\xi^t, \phi_\xi^t(N-1), \xi) - \nabla_\phi J_\mathcal{A}(\theta_\xi^t, \phi_\xi^t(N-1), \xi) + \nabla_\phi J_\mathcal{A}(\theta_\xi^t, \phi_\xi^t(N-1), \xi)\|^2 | \mathcal{F}_{N-1}^t]$$
$$= \mathbb{E}[\|(\hat{\nabla}_\phi - \nabla_\phi) J_\mathcal{A}(\theta_\xi^t, \phi_\xi^t(N-1), \xi)\|^2 | \mathcal{F}_{N-1}^t] + \mathbb{E}[\|\nabla_\phi J_\mathcal{A}(\theta_\xi^t, \phi_\xi^t(N-1), \xi)\|^2 | \mathcal{F}_{N-1}^t]$$
$$\quad + \mathbb{E}[2\langle (\hat{\nabla}_\phi - \nabla_\phi) J_\mathcal{A}(\theta_\xi^t, \phi_\xi^t(N-1), \xi), \nabla_\phi J_\mathcal{A}(\theta_\xi^t, \phi_\xi^t(N-1), \xi) \rangle | \mathcal{F}_{N-1}^t]$$
$$\leq \frac{\sigma^2}{N_b} + \|\nabla_\phi J_\mathcal{A}(\theta_\xi^t, \phi_\xi^t(N-1), \xi)\|^2.$$

Applying the PL condition (Assumption 3.4), and Assumption D.5 (a) we obtain

$$\mathbb{E}[V(\theta^t) - \ell_\mathcal{D}(\theta, \phi^t(N)) | \phi^{N-1}] - V(\theta^t) - \ell_\mathcal{D}(\theta, \phi^t(N-1))$$
$$\leq \mathbb{E}_\xi \left[ \frac{1}{L_{21}} \langle \nabla_\phi \mathcal{L}_\mathcal{D}, \nabla_\phi \mathcal{L}_\mathcal{A}(\theta^t, \phi_\xi^t(N-1), \xi) \rangle + \frac{L_{22}}{2L_{21}^2} (\frac{\sigma^2}{N_b} + \|\nabla_\phi \mathcal{L}_\mathcal{A}(\theta^t, \phi_\xi^t(N-1), \xi)\|^2) \right]$$
$$= \mathbb{E}_\xi \left[ -\frac{1}{2L_{22}} \|\nabla_\phi \mathcal{L}_\mathcal{D}\|^2 + \frac{1}{2L_{22}} \|\nabla_\phi (\mathcal{L}_\mathcal{D} + \frac{L_{22}}{L_{21}} \mathcal{L}_\mathcal{A})(\theta^t, \phi_\xi^t(N-1), \xi)\|^2 + \frac{L_{22}\sigma^2}{2L_{21}^2 N_b} \right]$$
$$\leq \frac{\mu}{cL_{21}} (\max_\phi \ell_\mathcal{D}(\theta^t, \phi) - \ell_\mathcal{D}(\theta^t, \phi^t(N-1))) + \frac{L_{22}\sigma^2}{2L_{21}^2 N_b},$$

rearranging the terms yields

$$\mathbb{E}[V(\theta^t) - \ell_{\mathcal{D}}(\theta^t, \phi^t(N))|\mathcal{F}_n^t] \leq \rho(V(\theta^t) - \ell_{\mathcal{D}}(\theta^t, \phi^t(N-1))) + \frac{L_{22}\sigma^2}{2L_{21}^2 N_b},$$

where we use the fact that $-\max_\phi \ell_{\mathcal{D}}(\theta^t, \phi) \leq -V(\theta^t)$. Telescoping the inequalities from $\tau = 0$ to $\tau = N$, we arrive at

$$\mathbb{E}[V(\theta^t) - \ell_{\mathcal{D}}(\theta^t, \phi^t(N))] \leq \rho^N(V(\theta^t) - \ell_{\mathcal{D}}(\theta^t, \phi^t(0))) + \frac{1-\rho^N}{1-\rho}\left(\frac{L_{22}\sigma^2}{2L_{21}^2 N_b}\right).$$

PL-condition implies quadratic growth, we also know that $V(\theta^t) - \ell_{\mathcal{D}}(\theta^t, \phi^t(N)) \leq \mathbb{E}_\xi \frac{1}{2\mu}\|\nabla_\phi \mathcal{L}_{\mathcal{D}}(\theta^t, \phi_\xi^t(N), \xi)\|^2 \leq \frac{1}{2\mu}G^2$, by Assumption 3.3,

$$\|\phi_\xi^*(\theta^t) - \phi_\xi^t(N)\|^2 \leq \frac{2}{\mu}(\mathcal{L}_{\mathcal{A}}(\theta^t, \phi_\xi^*, \xi) - \mathcal{L}_{\mathcal{A}}(\theta^t, \phi_\xi^t(N), \xi))$$

$$\leq \frac{2|c|}{\mu}|\mathcal{L}_{\mathcal{D}}(\theta^t, \phi_\xi^*, \xi) - \mathcal{L}_{\mathcal{D}}(\theta^t, \phi_\xi^t(N), \xi)|$$

Hence, with Jensen inequality and choice of $N_{\mathcal{A}}$ and $N_b$,

$$\mathbb{E}[\|z_t\|] = \mathbb{E}[\|\nabla_\theta V(\theta^t) - \mathbb{E}_\xi \nabla_\theta \mathcal{L}_{\mathcal{D}}(\theta^t, \phi_\xi^t(N_{\mathcal{A}}), \xi)\|]$$

$$\leq L_{12}\mathbb{E}[\|\phi_\xi^t(N_{\mathcal{A}}) - \phi_\xi^*\|]$$

$$\leq L_{12}\sqrt{\frac{2|c|}{\mu}\mathbb{E}[V(\theta^t) - \ell_{\mathcal{D}}(\theta^t, \phi^t(N_{\mathcal{A}}))]}$$

$$\leq L_{12}\sqrt{\frac{|c|}{\mu^2}\rho^{N_{\mathcal{A}}}G^2 + (1-\rho^{N_{\mathcal{A}}})\frac{|c|L_{22}^2\sigma^2}{\mu L_{21}^2 N_b}}.$$

Now we adjust the size of $N_{\mathcal{A}}$ and $N_b$ to make $\mathbb{E}[\|z_t\|]$ small enough, to this end, we set

$$\rho^{N_{\mathcal{A}}}\frac{|c|G^2}{\mu^2} \leq \frac{\varepsilon^4 L^2}{32D_V^2(2V_\infty + LD_\Theta)^4 \bar{L}}$$

$$\frac{|c|L_{22}^2\sigma^2}{L_{21}^2 N_b} \leq \frac{\varepsilon^4 L^2\mu^2}{32D_V^2(2V_\infty + LD_\Theta)^4 \bar{L}},$$

which further indicates that

$$N_{\mathcal{A}} \geq \frac{1}{\log \rho^{-1}}\log \frac{32D_V^2(2V_\infty + LD_\Theta)^4 \bar{L}|c|G^2}{L^2\mu^2\varepsilon^4}$$

$$N_b \geq \frac{32\mu L_{21}^2 D_V^2(2V_\infty + LD_\Theta)^4}{|c|L_{22}^2\sigma^2 \bar{L}L\varepsilon^4}.$$

In the setting above, it is not hard to verify that

$$\mathbb{E}[\|z_t\|] \leq \frac{L\varepsilon^2}{4D_V(2V_\infty + LD_\Theta)^2} \leq \varepsilon.$$

Also note that $\|\nabla_\phi \mathcal{L}_{\mathcal{A}}(\theta^t, \phi_\xi^t(N_{\mathcal{A}}), \xi)\| = \|\nabla_\phi \mathcal{L}_{\mathcal{A}}(\theta^t, \phi_\xi^t(N_{\mathcal{A}}), \xi) - \nabla_\phi \mathcal{L}_{\mathcal{A}}(\theta^t, \phi_\xi^*, \xi)\|$, given the proper choice of $N_{\mathcal{A}}$ and $N_b$, one has

$$\mathbb{E}\|\nabla_\phi \mathcal{L}_{\mathcal{A}}(\theta^t, \phi_\xi^t(N_{\mathcal{A}}), \xi) - \nabla_\phi \mathcal{L}_{\mathcal{A}}(\theta^t, \phi_\xi^*, \xi)\|$$

$$\leq L_{21}\mathbb{E}[\|\phi_\xi^t(N_{\mathcal{A}}) - \phi_\xi^*\|] \leq \frac{L\varepsilon^2}{4D_V(2V_\infty + LD_\Theta)^2} \leq \varepsilon,$$

which indicates the $\xi$-wise inner loop stability. $\qquad\square$

Now we are ready to provide the convergence guarantee of the first-order outer loop.

**Theorem D.7.** *Under Assumption D.5, Assumption 3.3, and Assumption D.2, let the stepsizes be,* $\kappa_{\mathcal{A}} = \frac{1}{L_{22}}$, $\kappa_{\mathcal{D}} = \frac{1}{L}$, *if* $N_{\mathcal{D}}$, $N_{\mathcal{A}}$, *and* $N_b$ *are large enough,*

$$N_{\mathcal{D}} \geq N_{\mathcal{D}}(\varepsilon) \sim \mathcal{O}(\varepsilon^{-2}) \quad N_{\mathcal{A}} \geq N_{\mathcal{A}}(\varepsilon) \sim \mathcal{O}(\log \varepsilon^{-1}), \quad N_b \geq N_b(\varepsilon) \sim \mathcal{O}(\varepsilon^{-4})$$

*then there exists* $t \in \mathbb{N}$ *such that* $(\theta^t, \{\phi_{\xi}^t(N_{\mathcal{A}})\}_{\xi \in \Xi})$ *is* $\varepsilon$-*meta-FOSE.*

*Proof.* According to the update rule of the outer loop, (here we omit the projection analysis for simplicity)

$$\theta^{t+1} - \theta^t = \frac{1}{L}\hat{\nabla}_\theta \ell_{\mathcal{D}}(\theta^t, \phi^t(N_{\mathcal{A}})),$$

one has, due to unbiasedness assumption, let $(\mathcal{F}_t)_{0 \leq t \leq N_{\mathcal{D}}}$ be the filtration generated by $\sigma(\theta^t | k \leq t)$

$$\mathbb{E}[\langle \nabla_\theta \ell_{\mathcal{D}}(\theta^t, \phi^t(N_{\mathcal{A}})), \theta^{t+1} - \theta^t \rangle | \mathcal{F}_t] = \frac{1}{L}\mathbb{E}[\|\nabla_\theta \ell_{\mathcal{D}}(\theta^t, \phi^t(N_{\mathcal{A}}))\|^2 | \mathcal{F}_t]$$
$$= L\mathbb{E}\|\theta^{t+1} - \theta^t\|^2 | \mathcal{F}_t],$$

which leads to

$$\mathbb{E}[\langle \nabla_\theta \ell_{\mathcal{D}}(\theta^t, \phi^*), \theta^{t+1} - \theta^t \rangle | \mathcal{F}_t] = \mathbb{E}[\langle z_t, \theta^t - \theta^{t+1} \rangle | \mathcal{F}_t] + L\mathbb{E}[\|\theta^{t+1} - \theta^t\|^2\|].$$

Since $V(\cdot)$ is $L$-Lipschitz smooth,

$$\mathbb{E}[V(\theta^t) - V(\theta^{t+1})] \leq \mathbb{E}[\langle \nabla_\theta V(\theta^t), \theta^t - \theta^{t+1} \rangle] + \frac{L}{2}\mathbb{E}[\|\theta^{t+1} - \theta^t\|^2]$$

$$\leq \mathbb{E}[\langle z_t, \theta^{t+1} - \theta^t \rangle] - \mathbb{E}[\langle \nabla_\theta \ell_{\mathcal{D}}(\theta^t, \phi^t(N_{\mathcal{A}})), \theta^{t+1} - \theta^t \rangle] + \frac{L}{2}\mathbb{E}[\|\theta^{t+1} - \theta^t\|^2] \quad \text{(D8)}$$

$$\leq \mathbb{E}[\langle z_t, \theta^{t+1} - \theta^t \rangle] - \frac{L}{2}\mathbb{E}[\|\theta^{t+1} - \theta^t\|^2].$$

Fixing a $\theta \in \Theta$, let $e_t := \langle \nabla_\theta \ell_{\mathcal{D}}(\theta^t, \phi^t(N_{\mathcal{A}})), \theta - \theta^t \rangle$, we have

$$\mathbb{E}[e_t | \mathcal{F}_t] = L\mathbb{E}[\langle \theta^{t+1} - \theta^t, \theta - \theta^t \rangle | \mathcal{F}_t]$$
$$= \mathbb{E}[\langle \nabla_\theta \ell_{\mathcal{D}}(\theta^t, \phi^t(N_{\mathcal{A}})) - \nabla_\theta V(\theta^t), \theta^{t+1} - \theta^t \rangle + \langle \nabla_\theta V(\theta^t), \theta^{t+1} - \theta^t \rangle]$$
$$\quad + L\mathbb{E}[\langle \theta^{t+1} - \theta^t, \theta - \theta^{t+1} \rangle] \quad \text{(D9)}$$
$$\leq \mathbb{E}[(\|z_t\| + V_\infty + LD_\Theta)\|\theta^{t+1} - \theta^t\|]$$

By the choice of $N_b$, we have, since $V_\infty = \max\{\max_\theta \|\nabla V(\theta)\|, 1\}$,

$$\mathbb{E}[\|z_t\|] \leq L_{12}\mathbb{E}[\|\phi^N - \phi^*\|] \leq \frac{L\varepsilon^2}{4D_V(2V_\infty + LD_\Theta)} \leq V_\infty.$$

Thus, the relation equation D9 can be reduced to

$$\mathbb{E}[e_t] \leq (2V_\infty + LD_\Theta)\mathbb{E}[\|\theta^{t+1} - \theta^t\|].$$

Telescoping equation D8 yields

$$-D_V \leq \mathbb{E}[V(\theta^0) - V(\theta^{N_{\mathcal{D}}})] \leq D_\Theta \sum_{t=0}^{T-1} \mathbb{E}[\|z_t\|] - \frac{L}{2(2V_\infty + LD_\Theta)^2}\mathbb{E}[\sum_{t=0}^{T-1} \mathbb{E}[e_t^2 | \mathcal{F}_t].$$

Thus, setting $N_{\mathcal{D}} \geq \frac{4D_V(2V_\infty + LD_\Theta)^2}{L\varepsilon^2}$, and then by Lemma D.6, we obtain that,

$$\frac{1}{N_{\mathcal{D}}}\sum_{t=0}^{N_{\mathcal{D}}-1} \mathbb{E}[e_t^2] \leq \frac{\varepsilon^2}{2} + \frac{2D_V(2V_\infty + LD_\Theta)^2}{LN_{\mathcal{D}}} \leq \varepsilon^2$$

which implies there exists $t \in \{0, \ldots, N_{\mathcal{D}} - 1\}$ such that $\mathbb{E}[e_t^2] \leq \varepsilon^2$.

$\square$

### D.3 Generalization to Unseen Attacks

In this section, we give a more concrete presentation of Proposition 2.2. Our main goal is to quantify the value discrepancy under an attack type that is out of empirical distribution. We consider attack types $\xi_1, \ldots, \xi_m$ to be empirically sampled from distribution $Q(\cdot)$ during the pre-training stage, and an unseen attack type $\xi_{m+1}$ in the online stage. The quantification of distance $C(\xi_{m+1}, \{\xi_i\}_{i=1}^m)$ relies on the total variation,

**Definition D.8** (total variation). For two distributions $P$ and $Q$, defined over the sample space $\Omega$ and $\sigma$-field $\mathcal{F}$, the total variation between $P$ and $Q$ is $\|P - Q\|_{TV} := \sup_{U \in \mathcal{F}} |P(U) - Q(U)|$.

The celebrated result shows the following characterization of total variation,

$$\|P - Q\|_{TV} = \sup_{f:0 \le f \le 1} \mathbb{E}_{x \sim P}[f(x)] - \mathbb{E}_{x \sim Q}[f(x)].$$

Let the fixed attack policies $\phi_i$, $i = 1, \ldots, m+1$ corresponding to each attack type. To formalize the generalization error, for each $\theta \in \Theta$, we define populational values

$$\hat{V}(\theta) := \frac{1}{m} \sum_{i=1}^m \mathbb{E}_{\tau \sim q_i^\theta} J_\mathcal{D}(\theta - \eta \hat{\nabla}_\theta J_\mathcal{D}(\tau), \phi_i, \xi_i)$$

$$\hat{V}_{m+1}(\theta) := \mathbb{E}_{\tau \sim q_{m+1}^\theta} J_\mathcal{D}(\theta - \eta \hat{\nabla}_\theta J_\mathcal{D}(\tau), \phi_{m+1}, \xi_{m+1})$$

where $q_i^\theta(\cdot) : (S \times A \times S)^{H-1} \times S \to [0, 1]$ is the trajectory distribution determined by state dependent policies $\pi_\mathcal{D}(\cdot|s; \theta)$, $\pi_\mathcal{A}(\cdot|s; \phi_i, \xi_i)$ and transition kernel $\mathcal{T}$. Since $q_i^\theta$ is factorizable, we have Lemma D.9 to eliminate $\|q_i^\theta - q_{m+1}^\theta\|_{TV}$ dependence on $\theta$ by upper bounding it using another pair of mariginal distributions.

**Lemma D.9.** *For any $\theta \in \Theta$, there exist marginals $d_i, d_{m+1} : (S \times A_\mathcal{A} \times S)^{H-1} \times S \to [0, 1]$ total variation $\|q_i^\theta - q_{m+1}^\theta\|_{TV}$ can be bounded by $\|d_i - d_{m+1}\|_{TV}$.*

*Proof.* By factorization, for a trajectory $\tau$, any $\theta \in \Theta$, and any type index $i = 1, \ldots, m+1$:

$$q_i^\theta(\tau) = \prod_{t=1}^{H-1} \pi_\mathcal{D}(a_\mathcal{D}^t|s_t; \theta) \prod_{t=1}^{H-1} \pi_\mathcal{A}(a_\mathcal{A}^t|s_t, \phi_i, \xi_i) \prod_{t=1}^{H-1} \mathcal{T}(s_{t+1}|s_t, a_t),$$

thus, by the inequality of product measure,

$$\|q_i^\theta - q_{m+1}^\theta\|_{TV} \le \sum_{t=1}^{H-1} \underbrace{\|\pi_\mathcal{D}(\cdot|s_t; \theta) - \pi_\mathcal{D}(\cdot|s_t; \theta)\|_{TV}}_{0} + \|d_i - d_{m+1}\|_{TV},$$

where $d_i$ and $d_{m+1}$ are the residue factors after removing $\pi_\mathcal{A}(\cdot|s_t; \theta)$. $\qquad\square$

**Assumption D.10.** For any $\xi \in \Xi$ and $\phi_\xi$, the function $J_\mathcal{D}(\theta, \phi_\xi, \xi)$ is $G$-Lipschitz continuous w.r.t. $\theta \in \Theta$;

**Proposition D.11.** *Under assumption 3.4 and certain regularity conditions, fixing a policy $\theta \in \Theta$, we have, there exist some marginal distribution of*

$$|\hat{V}_{m+1}(\theta) - \hat{V}(\theta)| \le C(d_{m+1}, \{d_i\}_{i=1}^m),$$

*where the constant $C$ depending on the total variation between $d_{m+1}$ and $\{d_i\}_{i=1}^m$:*

$$C(d_{m+1}, \{d_i\}_{i=1}^m) := \frac{2\eta G^2}{m} \sum_{i=1}^m \|d_{m+1} - d_i\|_{TV} + \frac{1 - \gamma^H}{1 - \gamma} \|d_{m+1} - \frac{1}{m} \sum_{i=1}^m d_i\|_{TV},$$

*here, $G$ is the Lipschitz parameter of $J_\mathcal{D}$ w.r.t. both $\theta$.*

*Proof.* We start with the decomposition of the generalization error, for an arbitrary attack type $\xi_i$, $i = 1, \ldots, m$, fixing a policy $\theta \in \Theta$ determines jointly with each $\phi_i$ the trajectory distribution $q_i^\theta$.

Denoting the one-step adaptation policy $\theta'(\tau) = \theta - \eta\nabla J_{\mathcal{D}}(\tau)$ as a function of trajectory $\tau$, we have the following decomposition,

$$\hat{V}_{m+1}(\theta) - \hat{V}(\theta) = \mathbb{E}_{\tau_{m+1}\sim q^{\theta}_{m+1}} J_{\mathcal{D}}(\theta'(\tau_{m+1}), \phi_{m+1}, \xi_{m+1}) - \frac{1}{m}\sum_{i=1}^{m}\mathbb{E}_{\tau_i\sim q^{\theta}_i} J_{\mathcal{D}}(\theta'(\tau_i), \phi_i, \xi_i)$$

$$= \underbrace{\mathbb{E}_{\tau_{m+1}\sim q^{\theta}_{m+1}} J_{\mathcal{D}}(\theta'(\tau_{m+1}), \phi_{m+1}, \xi_{m+1}) - \frac{1}{m}\sum_{i=1}^{m}\mathbb{E}_{\tau_{m+1}\sim q^{\theta}_{m+1}} J_{\mathcal{D}}(\theta'(\tau_{m+1}), \phi_i, \xi_i)}_{(i)}$$

$$+ \underbrace{\frac{1}{m}\sum_{i=1}^{m}\mathbb{E}_{\tau_{m+1}\sim q^{\theta}_{m+1}} J_{\mathcal{D}}(\theta'(\tau_{m+1}), \phi_i, \xi_i) - \frac{1}{m}\sum_{i=1}^{m}\mathbb{E}_{\tau_i\sim q^{\theta}_i} J_{\mathcal{D}}(\theta'(\tau_i), \phi_i, \xi_i)}_{(ii)}.$$

We assume $(\tau_{m+1}, \tau_i)$ is drawn from a joint distribution which has marginals $q^{\theta}_{m+1}$ and $q^{\theta}_i$ and is corresponding to the maximal coupling of these two. Then,

$$\tau_{m+1} \sim q^{\theta}_{m+1}, \quad \tau_i \sim q^{\theta}_i, \quad \mathbb{P}(\tau_{m+1} \neq \tau_i) = \|q^{\theta}_i - q^{\theta}_{m+1}\|_{TV},$$

if $\tau_{m+1}$ disagrees with $\tau_i$, for $(ii)$, we have, since $J^{\theta}_{\mathcal{D}}$ is Lipschitz with respect to $\theta$,

$$\|J_{\mathcal{D}}(\theta'(\tau_{m+1}), \phi_i, \xi_i) - J_{\mathcal{D}}(\theta'(\tau_i), \phi_i, \xi_i)\|$$
$$\leq \eta G\|\hat{\nabla}_{\theta} J_{\mathcal{D}}(\tau_{m+1}) - \hat{\nabla}_{\theta} J_{\mathcal{D}}(\tau_i)\|$$
$$\leq 2\eta G^2,$$

as a result, denoting the maximal coupling of $q^{\theta}_{m+1}$ and $q^{\theta}_i$ as gives,

$$[\mathbb{E}_{\tau_{m+1}\sim q^{\theta}_{m+1}} J_{\mathcal{D}}(\theta'(\tau_{m+1}), \phi_i, \xi_i) - \mathbb{E}_{\tau_i\sim q^{\theta}_i} J_{\mathcal{D}}(\theta'(\tau_i), \phi, \xi_i)]$$
$$= \mathbb{E}_{(\tau_{m+1}, \tau_i)\sim\prod(q^{\theta}_{m+1}, q^{\theta}_i)}[J_{\mathcal{D}}(\theta'(\tau_{m+1}), \phi_i, \xi_i) - J_{\mathcal{D}}(\theta'(\tau_i), \phi, \xi_i)]$$
$$\leq 2\eta G^2\|q^{\theta}_{m+1} - q^{\theta}_i\|_{TV} \leq 2\eta G^2\|d_i - d_{m+1}\|_{TV},$$

where the last inequality is due to Lemma D.9. Averaging the $m$ empirical $\xi_i$'s yeilds the result:

$$(ii) \leq \frac{2\eta G^2}{m}\sum_{i=1}^{m}\|d_i - d_{m+1}\|_{TV}.$$

Since the trajectory distribution is a product measure, the difference between $q^{\theta}_i$ and $q^{\theta}_{m+1}$ only lies by attacker's type, $\|q^{\theta'(\tau_{m+1})}_{m+1} - q^{\theta'(\tau_{m+1})}_i\|_{TV} = \|q^{\theta}_{m+1} - q^{\theta}_i\|_{TV} \leq \|d_{m+1} - d_i\|_{TV}$.

Now we bound $(i)$, for ease of exposition we let $q'' = q^{\theta'(\tau_{m+1})}_{m+1}$ and $q'_i := q^{\theta'(\tau_{m+1})}_i$. By the finiteness of total trajectory reward $R(\tau)$ for any trajectory $\tau$, $R(\tau) \leq \frac{1-\gamma^H}{1-\gamma}$, hence,

$$(i) = \mathbb{E}_{\tau_{m+1}\sim q^{\theta}_{m+1}} J_{\mathcal{D}}(\theta'(\tau_{m+1}), \phi_{m+1}, \xi_{m+1}) - \frac{1}{m}\sum_{i=1}^{m}\mathbb{E}_{\tau_{m+1}\sim q^{\theta}_{m+1}} J_{\mathcal{D}}(\theta'(\tau_{m+1}), \phi_i, \xi_i)$$

$$= \mathbb{E}_{\tau_{m+1}\sim q^{\theta}_{m+1}}\left[\mathbb{E}_{\tau''\sim q''} R_{\mathcal{D}}(\tau'') - \frac{1}{m}\sum_{i=1}^{m}\mathbb{E}_{\tau'_i\sim q'_i} R_{\mathcal{D}}(\tau'_i)\right]$$

$$\leq \mathbb{E}_{\tau_{m+1}\sim q^{\theta}_{m+1}} \frac{1-\gamma^H}{1-\gamma}\|q''_{m+1} - \frac{1}{m}\sum_{i=1}^{m} q'_i\|_{TV}$$

$$\leq \frac{1-\gamma^H}{1-\gamma}\|d_{m+1} - \frac{1}{m}\sum_{i=1}^{m} d_i\|_{TV}.$$

$$\square$$

# E EXPERIMENT SETUP

**Datasets.** We consider two datasets: MNIST (LeCun et al., 1998) and CIFAR-10 (Krizhevsky et al., 2009), and default $i.i.d.$ local data distributions, where we randomly split each dataset into $n$ groups, each with the same number of training samples. MNIST includes 60,000 training examples and 10,000 testing examples, where each example is a $28 \times 28$ grayscale image, associated with a label from 10 classes. CIFAR-10 consists of 60,000 color images in 10 classes of which there are 50,000 training examples and 10,000 testing examples. For the *non-i.i.d.* setting (see Figure 9(d)), we follow the method of (Fang et al., 2020) to quantify the heterogeneity of the data. We split the workers into $C = 10$ (for both MNIST and CIFAR-10) groups and model the *non-i.i.d.* federated learning by assigning a training instance with label $c$ to the $c$-th group with probability $q$ and to all the groups with probability $1 - q$. A higher $q$ indicates a higher level of heterogeneity.

**Federated Learning Setting.** We use the following default parameters for the FL environment: local minibatch size = 128, local iteration number = 1, learning rate = 0.05, number of workers = 100, number of backdoor attackers = 5, number of untargeted model poisoning attackers = 20, subsampling rate = $10\%$, and the number of FL training rounds = 500 (resp. 1000) for MNIST (resp. CIFAR-10). For MNIST, we train a neural network classifier of 8×8, 6×6, and 5×5 convolutional filter layers with ReLU activations followed by a fully connected layer and softmax output. For CIFAR-10, we use the ResNet-18 model (He et al., 2016). We implement the FL model with PyTorch (Paszke et al., 2019) and run all the experiments on the same 2.30GHz Linux machine with 16GB NVIDIA Tesla P100 GPU. We use the cross-entropy loss as the default loss function and stochastic gradient descent (SGD) as the default optimizer. For all the experiments except Figures 9(c) and 9(d), we fix the initial model and random seeds of subsampling for fair comparisons. we apply Neural Cleanse (Wang et al., 2019b) to reverse engineer the backdoor trigger.

**Baselines.** We evaluate our defense method against various state-of-the-art attacks, including non-adaptive and adaptive untargeted model poison attacks (i.e., IPM Xie et al. (2020), LMP Fang et al. (2020), RL Li et al. (2022a)), as well as backdoor attacks (BFL Bagdasaryan et al. (2020) without model replacement, BRL Li et al. (2023), with tradeof'?
f parameter $\lambda = 0.5$, DBA (Xie et al., 2019) where each selected attacker randomly chooses a sub-trigger as shown in Figures 7, PGD attack (Wang et al., 2020) with a projection norm of 0.05), and a combination of both types. To establish the effectiveness of our defense, we compare it with several strong defense techniques. These baselines include defenses implemented during the training stage, such as Krum Blanchard et al. (2017), Clipping Median Yin et al. (2018); Sun et al. (2019); Li et al. (2022a) (with norm bound 1), FLTrust Cao et al. (2021) with 100 root data samples and bias $q = 0.5$, training stage CRFL Xie et al. (2021) with norm bound of 0.02 and noise level $1e - 3$ as well as post-training defenses like Neuron Clipping Wang et al. (2022) and Pruning Wu et al. (2020). We use the original clipping thresholds 7 in Wang et al. (2022) and set the default pruning number to 256.

**Reinforcement Learning Setting.** In our RL-based defense, since both the action space and state space are continuous, we choose the state-of-the-art Twin Delayed DDPG (TD3) (Fujimoto et al., 2018) algorithm to individually train the untargeted defense policy and the backdoor defense policy. We implement our simulated environment with OpenAI Gym (Brockman et al., 2016) and adopt OpenAI Stable Baseline3 (Raffin et al., 2021) to implement TD3. The RL training parameters are described as follows: the number of FL rounds = 300 rounds, policy learning rate = 0.001, the policy model is MultiInput Policy, batch size = 256, and $\gamma = 0.99$ for updating the target networks. The default $\lambda = 0.5$ when calculating the backdoor rewards.

**Meta-learning Setting.** The attack domains (i.e., potential attack sets) are built as following: For meta-RL, we consider IPM (Xie et al., 2020), LMP (Fang et al., 2020), EB (Bhagoji et al., 2019) as three possible attack types. For meta-SG against untargeted model poisoning attack, we consider RL-based attacks (Li et al., 2022a) trained against Krum (Blanchard et al., 2017) and Clipping Median (Li et al., 2022a; Yin et al., 2018; Sun et al., 2019) as initial attacks. For meta-SG against backdoor attack, we consider RL-based backdoor attacks Li et al. (2023) trained against Norm-bounding (Sun et al., 2019) and Neuron Clipping (Wang et al., 2022) (Pruning Wu et al. (2020)) as initial attacks. For meta-SG against mix type of attacks, we consider both RL-based attacks (Li et al., 2022a) and RL-based backdoor attacks Li et al. (2023) described above as initial attacks.

At the pre-training stage, we set the number of iterations $T = 100$. In each iteration, we uniformly sample $K = 10$ attacks from the attack type domain (see Algorithm 2 and Algorithm 1). For each

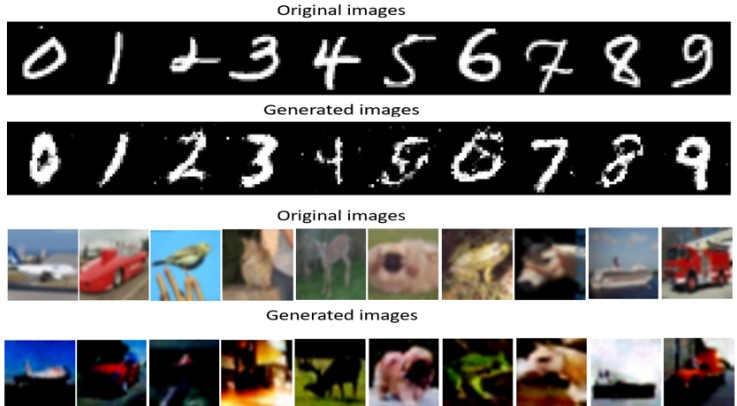

Figure 5: Self-generated MNIST images using conditional GAN Mirza & Osindero (2014) (second row) and CIFAR-10 images using a diffusion model Sohl-Dickstein et al. (2015) (fourth row).

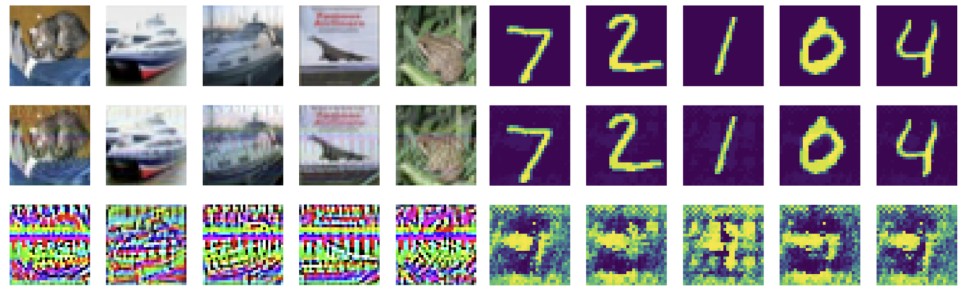

Figure 6: Generated backdoor triggers using GAN-based models Doan et al. (2021). Original image (first row). Backdoor image (second row). Residual (third row).

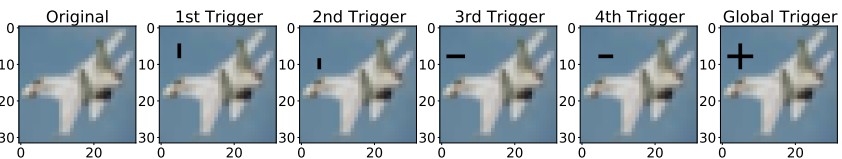

Figure 7: CIFAR-10 fixed backdoor trigger patterns.The global trigger is considered the default poison pattern and is used for online adaptation stage backdoor accuracy evaluation. The sub-triggers are used by pre-training and DBA only.

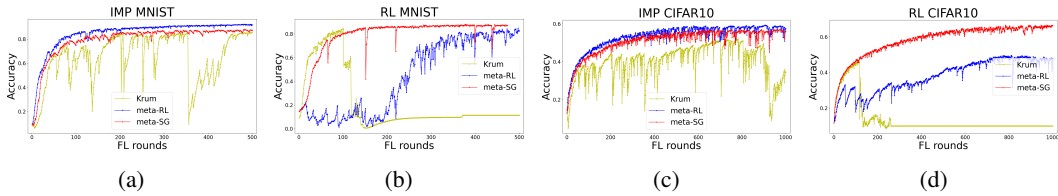

Figure 8: Comparisons of defenses against untargeted model poisoning attacks (i.e., IPM and RL) on MNIST and CIFAR-10. RL-based attacks are trained before epoch 0 against the associate defenses (i.e., Krum and meta-policy of meta-RL/meta-SG). All parameters are set as default.

attack, we generate a trajectory of length $H = 200$ for MNIST ($H = 500$ for CIFAR-10), and update both attacker's and defender's policies for 10 steps using TD3 (i.e., $l = N_\mathcal{A} = N_\mathcal{D} = 10$). At the online adaptation stage, the meta-policy is adapted for $100$ steps using TD3 with $T = 10$, $H = 100$ for MNIST ($H = 200$ for CIFAR-10), and $l = 10$. Other parameters are described as follows: single task step size $\kappa = \kappa_\mathcal{A} = \kappa_\mathcal{D} = 0.001$, meta-optimization step size $= 1$, adaptation step size $= 0.01$.

**Space Compression.** Following the BSMG model, it is most generally to use $w_g^t$ or $(w_g^t, \mathbf{I}^t)$ as the state, and $\{\tilde{g}_k^t\}_{k=1}^{M_1+M_2}$ or $w_g^{t+1}$ as the action for the attacker and the defender, respectively, if the federated learning model is small. However, when we use federated learning to train a high-dimensional model (i.e., a large neural network), the original state/action space will lead to an extremely large search space that is prohibitive in terms of training time and memory space. We adopt the RL-based attack in (Li et al., 2022a) to simulate an adaptive model poisoning attack and the RL-based local search in (Li et al., 2023) to simulate an adaptive backdoor attack, both having a 3-dimensioanl real action spaces after action comparison. We further restrict all malicious devices controlled by the same attacker to take the same action. To compress the state space, we reduce $w_g^t$ to only include its last two hidden layers for both attacker and defender and reduce $\mathbf{I}^t$ to the number of malicious clients sampled at round $t$.

**Self-generated Data.** We begin by acknowledging that the server only holds a small amount of initial data (200 samples with $q = 0.1$ in this work) learned from first 20 FL rounds using inverting gradient Geiping et al. (2020), to simulate training set with 60,000 images (for both MNIST and CIFAR-10) for FL. This limited data is augmented using several techniques such as normalization, random rotation, and color jittering to create a larger and more varied dataset, which will be used as an input for generative models.

For MNIST, we use the augmented dataset to train a Conditional Generative Adversarial Network (cGAN) model Mirza & Osindero (2014); Odena et al. (2017) built upon the codebase in Lacerda (2018). The cGAN model for the MNIST dataset comprises two main components - a generator and a discriminator, both of which are neural networks. Specifically, we use a dataset with 5,000 augmented data as the input to train cGAN, keep the network parameters as default, and set the training epoch as 100.

For CIFAR-10, we leverage a diffusion model implemented in Crowson (2018) that integrates several recent techniques, including a Denoising Diffusion Probabilistic Model (DDPM) Ho et al. (2020), DDIM-style deterministic sampling Song et al. (2020), continuous timesteps parameterized by the log SNR at each timestep Kingma et al. (2021) to enable different noise schedules during sampling. The model also employs the 'v' objective, derived from Progressive Distillation for Fast Sampling of Diffusion Models Salimans & Ho (2022), enhancing the conditioning of denoised images at high noise levels. During the training process, we use a dataset with 50,000 augmented data samples as the input to train this model, keep the parameters as default, and set the training epoch as 30.

**Backdoor Attacks.** We consider the trigger patterns shown in Figure 6 and Figure 7 for backdoor attacks. For triggers generated using GAN (see Figure 6), the goal is to classify all images of different classes to the same target class (all-to-one). For fixed patterns (see Figure 7), the goal is to classify images of the airplane class to the truck class (one-to-one). The default poison ratio is 0.5 in both cases. The global trigger in Figure 7 is considered the default poison pattern and is used for the online adaptation stage for backdoor accuracy evaluation.

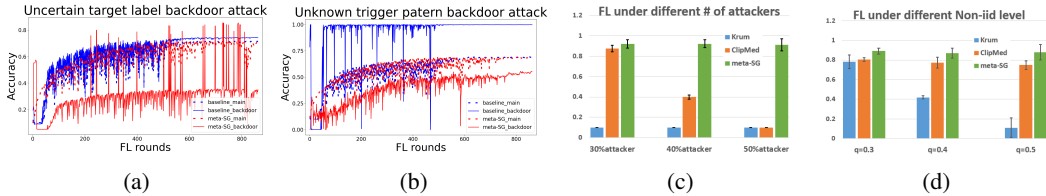

(a)             (b)             (c)             (d)

Figure 9: Ablation studies. (a)-(b): uncertain backdoor target and unknown backdoor triggers, where the meta-policies are trained by distributions of triggers generated by GAN-based models Doan et al. (2021) targeting multiple labels on CIFAR-10. The baseline defense combines the training-stage norm bounding with bound 2 and the post-training Neuron Clipping with clip range 7. (c)-(d): meta-SG trained by the number of malicious clients in $[40, 50, 60]$ and non-$i.i.d.$ level in $q = [0.5, 0.6, 0.7]$ on MNIST compared with Krum and Clipping Median under the known LMP attack. Other parameters are set as default.

## F    ADDITIONAL EXPERIMENT RESULTS

**More untargetd model poisoning results.** Similar to results in Figure 3 as described in Section 4, meta-RL achieves the best performance (slightly better than meta-SG) under IPM attacks for both MNIST and CIFAR-10. On the other hand, meta-SG performs the best (significantly better than meta-RL) against RL-based attacks for both MNIST and CIFAR-10. Notably, Krum can be easily compromised by RL-based attacks by a large margin. In contrast, meta-RL gradually adapts to adaptive attacks, while meta-SG displays near-immunity against RL-based attacks.

**Blackbox backdoor defense.** In the ablation study, we exam the meta-policy of meta-SG trained by using reverse engineering targeting all 10 possible labels in CIFAR-10. Figure 9(a) shows the defense performance of the meta-policy against a GAN-based attack targeting label 0 in the real FL environment. Although meta-SG reduces the backdoor accuracy by a large amount (nearly two-thirds), its performance is unstable due to the fact that the meta-policy will occasionally target a wrong label, even with adaptation. In Figure 9(b), the meta-policy is trained using GAN-based BRL attacks and tested against the BRL attack with a fixed global pattern Li et al. (2023) (see Appendix E for details). Even though the defense performance of meta-SG against backdoor attack is significantly better than the baseline, the backdoor accuracy still reaches nearly $50\%$ at the end of FL training.

**Number of malicious clients/Non-i.i.d. level.** Here we apply our meta-SG framework to study the impact of inaccurate knowledge of the number of malicious clients and the non-$i.i.d.$ level of clients' local data distribution. With rough knowledge that the number of malicious clients is in the range of 5-60, we consider three cases with 40, 50, and 60 malicious clients, respectively, during meta-learning. Similarly, we assume the non-$i.i.d.$ level is between 0.1-0.7 and consider q= $0.5, 0.6, 0.7$ during meta-learning. As illustrated in Figures 9(c) and 9(d), meta-SG reaches the highest model accuracy for all numbers of malicious clients and non-$i.i.d.$ levels under LMP, where the attack type is known to the defender.

## G    BROADER IMPACTS AND LIMITATIONS

**Meta Equilibrium and Information Asymmetry.** Information asymmetry is a prevailing phenomenon arising in a variety of contexts, including adversarial machine learning (e.g. FL discussed in this work), cyber security (Manshaei et al., 2013), and large-scale network systems (Li et al., 2022c). Our proposed meta-equilibrium (Definition 2.1) offers a data-driven approach tackling asymmetric information structure in dynamic games without Bayesian-posterior beliefs. Achieving the strategic adaptation through stochastic gradient descent, the meta-equilibrium is computationally superior to perfect Bayesian equilibrium and better suited for real-world engineering systems involving high-dimensional continuous parameter spaces. It is expected that the meta-equilibrium can also be relevant to other adversarial learning contexts, cyber defense, and decentralized network systems.

**First-order Method with Strict Competitiveness.** Due to the hardness of the stochastic bilevel optimization problem, we have expanded our search scope with an alternative solution concept that merely involves the first-order necessary conditions for meta-SE. Our analytical result relies on the special game structure induced by the strict competitiveness assumption, which essentially "aligns" the defender/attacker objectives leveraging the nature of policy gradient, despite them being general-sum. Relaxing this assumption allows our framework to deal with a more general class of problems, yet may potentially disrupt the Danskin-type structure of gradient estimation. For simplicity of

exposition, we neglected the stochastic analysis for the defender policy gradient estimation in the outer loop of the algorithm, the concentration of which depends on the trajectory batch size and attacker-type sample size. We leave the outer loop sample-complexity analysis to future work.

**Incomplete Universal Defense.** Our aim is to establish a comprehensive framework for universal federated learning defense. This framework ensures that the server remains oblivious to any details pertaining to the environment or potential attackers. Still, it possesses the ability to swiftly adapt and respond to uncertain or unknown attackers during the actual federated learning process. Nevertheless, achieving this universal defense necessitates an extensive attack set through pre-training, which often results in a protracted convergence time toward a meta-policy. Moreover, the effectiveness and efficiency of generalizing from a wide range of diverse distributions pose additional challenges. Considering these, we confine our experiments in this paper to specifically address a subset of uncertainties and unknowns. This includes variables such as the type of attacker, the number of attackers, the level of independence and identically distributed data, backdoor triggers, backdoor targets, and other relevant aspects. However, we acknowledge that our focus is not all-encompassing, and there may be other factors that remain unexplored in our research.