# OpenReview forum: "Towards Universal Robust Federated Learning via Meta Stackelberg Game"
_ICLR.cc/2024/Conference — Submitted to ICLR 2024_

### Official Review · Reviewer_UDsm · 2023-10-31

**Soundness:** 2 fair
**Presentation:** 1 poor
**Contribution:** 1 poor
**Rating:** 3
**Confidence:** 3

**Summary:**

This paper proposes a pre-training algorithm to defend against multiple attacks in the context of federated learning. In particular, it considers the threat model in which adversaries can either use a model poisoning attack that aims to maximize average loss, or use a backdoor attack that aims to cause misclassification of poisoned test data while preserving decent performance on clean test data. The idea of the proposed algorithm is to converge towards an equilibrium where the central server has previously learned to defend itself against the mentioned attacks, and then learns in the context of the federated learning setting. The authors propose both a theoretical analysis of the number of gradient iterations and an experimental evaluation of the proposed solution.

**Strengths:**

- The game theoretic approach is interesting.
- The experimental results show the improvement of the proposed method over some existing defenses.

**Weaknesses:**

In my opinion, the form of the paper is not presentable and does not meet ICLR standards (the following points are not sorted by importance):

- Page 2 in FL process: learning rate is missing in the gradient descent formula

- Definition 2.1: As it is presented, there isn’t any condition for $\theta$ and $\phi$ to constitute an equilibrium. More precisely, if I had to simplify the current formulation of the definition, one would have:
"Definition 2.1 :$\theta$ and $\phi$ constitute an equilibrium if they satisfy : $\max_{\theta} f(\theta)$"
For me there is definitely a problem in the formulation.

- Figure 2 shows that the pre-training (that corresponds to Algorithm 1) output is the policy $\pi_\theta$ and the gradient adaptation $\Psi$, whereas in Algorithm 1, the output is $\theta$. This is a bit confusing.

- As presented in the paper, the objective of federated learning is to minimize the loss on all clients. I think it would be better to say (or to add) that the objective of federated learning under attack is to minimize the loss on the set of honest clients and not on all the clients, since some of them are malicious.

- In Definition 3.1, the intersection $\Uptheta \cap B(\theta^\star)$ in the first 'max' is exactly equal to $B(\theta^\star)$. Can you explain this choice? Same problem for the second 'max', and above all there is an error in the definition of $B(\phi^\star)$ which depends on theta, whereas the constraint is on $\phi$...

- The 'Meta_Update' function in Algorithm 1 is not explained anywhere.... Or maybe I just didn't find it.

- Most of the graphs in the experiments section are not visible at all... the font size is too small.

I found the paper very difficult to follow because of the points mentioned above.

On the content :

- The state-of-the-art in defending against model poisoning (a.k.a Byzantine attacks) is to use the NNM [1] pre-aggregation rule before using any aggregation rule such as Krum or Trimmed-Mean etc... Why didn't the authors use this technique to compare with the proposed method?

- Most of the time, the FL setting appears either when data needs to be kept on the client side, or when computing power needs to be divided. In both cases, it is assumed that the server will not learn on its own because it is not possible or practical to do so. Is there any practical reason why it is acceptable to consider that the central server is able to pre-train here? How does the server generate data? Overall, if the server is allowed to pre-train, would it not allow to train completely in a centralized way and avoid potential malicious client?

[1] Youssef Allouah, Sadegh Farhadkhani, Rachid Guerraoui, Nirupam Gupta, Rafaël Pinot, and John Stephan. Fixing by mixing: A recipe for optimal Byzantine ML under heterogeneity, 2023.

**Questions:**

See above

---

> ### Author Response · Authors · 2023-11-20
>
> We thank the reviewer for constructive comments.
>
> Q.1 Missing learning rate
>
> The learning rate is modeled in aggregation rule $\textit{Aggr}$.
>
> Q.2 Equilibrium definition
>
> There is an interdependency between the defender's and the attacker's equilibrium policy, which the maximization over $\theta$ fails to capture. To see this, we first note that the attacker's policy $\phi_\xi^*$ is dependent on the defender's policy $\theta$, i.e., $\phi_\xi^*=\phi_\xi^*(\theta)$. Then, with this dependency, the defender's problem is $\max_{\theta \in \Theta} \mathbb{E}_{\xi \sim Q} \mathbb{E}\tau [J_D(\theta+\eta\hat{\nabla}J_D(\tau), \phi\_\xi^*(\theta), \xi)]$.  Hence, the variable $\theta$ influences the objective in two ways: 1) directly as the defender's policy $\theta$, and 2) indirectly through the attacker's policy $\phi(\theta)$.  Using the reviewer's language, the equilibrium definition is equivalent to the following bilevel optimization
> \begin{align*}
>     \max\_{\theta\in \Theta} &\quad f(\theta, \phi^*(\theta))\\
>      \text{subject to} &\quad \phi^*(\theta)\in \arg\max g(\theta, \phi),
> \end{align*}
> where $f$ and $g$ correspond to the defender's and attacker's objective functions, respectively. This bilevel optimization is the standard formulation of the Stackelberg equilibrium.
>
> Q.3 The output of pre-training
>
> The gradient adaptation $\Psi$ is pre-fixed and does not belong to the meta-Stackelberg learning process. We add $\Psi$ to Figure 2 to highlight that the learned meta equilibrium policy $\pi_\theta$ is tied in with the gradient adaptation, and different adaptation mapping will lead to different meta policy in the pre-training. We will clarify this point in Figure 2 and Algorithm1.
>
> Q.4 FL loss over honest clients
>
> This definition of the FL loss is $F(w)=\frac{1}{n}\sum_{i=1}^n f(w, D_i)$, where $D_i$ is the $i$-th client's dataset, i.e., the global model is evaluated using the cleaning dataset. $F(w)$ is optimized over all the clients' data because, in practice, clients can be compromised temporarily and be "malicious," yet it can also be released afterward. More importantly, the server can not distinguish if a client is malicious and can not record the client ID in most cases.
>
> Q.5 Definition of $B(\theta)$ and $B(\phi)$
>
> We are sorry for the typo of $B(\phi^*)$. Yes, inside the brackets should be $\phi \in \Phi$ instead of $\theta \in \Theta$; the intersection in  $\Theta \bigcap B(\theta^*)$ and $ \Phi \bigcap B(\phi^*_\xi )$ is redundant and should be simply be $B(\theta^*)$ and $B(\phi^*_\xi )$. Ensuring $\|\theta - \theta^*\| \leq 1$ is to ensure the characterization of a local maximum. When  $B(\theta^*)$ retains the whole unit ball (every element in the ball is within $\Theta$), the condition here simply implies that the norms of the gradients are smaller than $\varepsilon$ in the unconstrained setting. To see this, we can simply put $\theta = \theta^* + \frac{\nabla_{\theta} \mathcal{L}\_\mathcal{D} (\theta^*, \phi^*\_\xi, \xi)}{\|\nabla\_{\theta} \mathcal{L}\_\mathcal{D} (\theta^*, \phi^*\_\xi, \xi)\|}$ and apply the condition.
>
> Q.6 Definition of Meta-Update
>
> Due to the page limit and the layout, the exact formulation is deferred to Appendix B, to which we provide the pointer at the end of page 5. Long story short, the meta update takes in all gradient estimates $\{\hat{\nabla} J\_\mathcal{D}(\xi)\}\_{\xi\in \hat{\Xi}}$ with respect to each sampled attack type. In our experiment, we use reptile-type meta update [1], and the resulting update scheme is to apply the average gradient: $\theta^{t+1}\gets \theta^t +1/K \sum\_{\xi\in \hat{\Xi}}\hat{\nabla} J\_\mathcal{D}(\xi)$. For theoretical analysis, we employ the debiased meta-update popularized by [2], please see Algorithm 3 in Appendix B for more details.
>
> Q.7 Figure not visible
>
> We will enlarge the font size in the latter version when the space is allowed.
>
> Q.8 Practicality of pre-training
>
> We will add comparisons and discussions when they release official codes. Instead of focusing on untargeted Byzantine attacks in NNM, our proposed method cares more about dealing with uncertain/unknown attacks, e.g., a mixture of untargeted and backdoor attacks with a potentially long-term objective. For single-type untargeted attacks under commonly used settings (20% attacker, non-adaptive attack), current baselines (e.g., Clipping Median, FLtrust) already perform well enough compared with FedAvg with no attack baseline in their papers. We will test more challenging settings in future works.
>
> [1] Nichol et al., On first-order meta-learning algorithms. arXiv preprint arXiv:1803.02999.
>
>
> [2] Fallah et al., On the convergence theory of debiased model-agnostic meta-reinforcement learning. Advances in Neural Information Processing Systems, 34, 3096-3107

---

### Official Review · Reviewer_D75P · 2023-11-01

**Soundness:** 3 good
**Presentation:** 2 fair
**Contribution:** 1 poor
**Rating:** 3
**Confidence:** 3

**Summary:**

The paper consider mitigation of Byzantine attacks in federated learning. They assume access to a simulator on which they find a minimax optimal policy for the defending aggregation policy in a pre-training phase. The pre-training proceeds by iteratively considering a batch of sampled attack type and subsequently unrolling a simulated training scenario using the current policy of both the defender and the attack type. An inner loop ensures that the attack policy is solved approximately optimally (at the expense of unrolling multiple times). A meta learning aggregation rule, such as Reptile, is used to aggregate the updates across the batch of sampled attack type. This allows adjusting to a particular (fixed and possibly unknown) attack type at training time more effectively. Convergence is shown under strict competition and PL conditions and the algorithm is demonstrated on MNIST and CIFAR10.

**Strengths:**

- The idea of a data dependent aggregation rule seems useful
- There is a good overview of existing literature

**Weaknesses:**

The scenario being modelled is a setting where the attacker randomly selects the attack type, fixes it throughout training and only optimizes the hyperparameters of the attack type. One major concern is that this is a very narrow scenario, which is subsequently solved by a (computationally expensive) heavy machinery. I would expect a simple baseline to do well. Why not in Table 1 compare against the same defences as used in the policy based method (e.g. FoolsGold)? How about comparing against a fixed policy (with reasonable defaults)?

One major issue is that the final algorithms is never fully specified since many parts remains undefined (even after looking through the appendix):

- What is an attacker type? Is it a fine set? Last paragraph of page 3 does not seem to define it precisely.
- What is the policy for a given attack type? In theory part is it a mapping from parameter space to parameter space? In practice is it the hyperparameters of a given attack type?
- Right before section 4.2:
    - You seem to optimize over $\mathbb R^3$ (with some additional constraints) for untargeted defences policy and similarly for backdoor defence. Should we understand that $\theta$ in Algorithm 1 lives in the _product_ space of the two?
    - Do you project to keep e.g. $b$ in $a_1^t:=(a,b,c)$ within the trimming threshold?
 - The online adaptation only seems to be described loosely in the paragraph right before section 3. Do you need to store a trajectory of model weights? What is the memory requirement?

 I suggest specifying the algorithm (as used in both theory and in experiments) in full detail.

Theory:

- The convergence results do not seem surprising or informative. All difficulty seems to be assumed away with PL conditions, strict-competitiveness, increasing batchsize and approximating a max-oracle. You also seem to be ignoring the size of the (sampled) attack type space. Is the batch size of the attack types taken "large enough"? (intuitively it should depend on $\varepsilon$)
- If the attack types space have no structure (it is a set) how can you give a meaningful OOD generalization bound? In the appendix it seems that you almost have to assume that their policies are not too different. Can you elaborate on what this proposition buys you?
- If the model is overparameterized the attacker can construct a backdoor attack without harming the defenders reward. How does solving (2) prevent a backdoor attack?

It seems that the methods intentionally _violates client privacy_ to construct the necessary simulation dataset. Bottom page 7 states "We use inference attack (i.e., Inverting gradient (Geiping et al., 2020)) in (Li et al., 2022a) for only a few FL epochs (20 in our setting) to learn data from clients". This does not seem viable. Do you have ablation over how much the simulated data quality degrades the performance?

**Questions:**

See the field above.

---

> ### Author Response · Authors · 2023-11-20
>
> We thank the reviewer for constructive feedback and comments.
>
> Q.1 Narrow scenarios
>
> we intend to build a general framework for FL to defend against uncertain/unknown attackers, which could be adaptive (i.e., learning-based attack), complex (i.e., mixed type attack), and potentially white-box (i.e., fully informed) attacker. Worst-case assumption is natural in security domain when the attack is unknown. To the best of our knowledge (see Figure 1), current defenses fail to defend such strong attacks. We allow the server to equip limited computational power to balance the trade-off of suffering huge loss among all clients under worst-case attacks. We consider three main categories of aggregation-based defenses, namely, client-wise (Krum), dimension-wise (Clipping Median), and vector-wise (FLtrust) in Table 1. FoolsGold uses cosine similarity to measure the distance between vectors, which is similar to FLtrust, but without root data. In our cases, we'd like to choose strong and representative baselines. Fixed policy defenses even with sophistic design fail to defend leaning-based, fully informed worst-case attack (e.g., RL-based attack) eventually, which motivate a learning-based defense. It is also hard to directly design a "fixed policy" against unseen attacks. How to find "reasonable defaults" is actually tricky here, since simple combination of current SOTA defenses will not work (see Figure 1(b)), thus, we propose RL to adaptively forge the combination of defenses.
>
>
> Q.2 Definition of attack types
>
>
> The attack type refers to the joint configuration of malicious clients, including the number of malicious clients employing the backdoor attacks $M_1$, the total number of malicious clients $M_2$, i.e., the number of clients employing untargeted attack is $M_2-M_1$, the backdoor attack method, denoted by $\texttt{BA}$, the untargeted model poisoning method, denoted by $\texttt{MP}$.  Mathematically, $\xi=(M_1, M_2, \texttt{BA}, \texttt{MP})$, where $\texttt{BA}$ and $\texttt{MP}$ are chosen from the attack domain $\{\texttt{EB}, \texttt{IPM}, \texttt{LMP}, \texttt{RL}, \texttt{BFL}, \texttt{DBA}, \texttt{PGD}, \texttt{BRL}\}$. The set of attack types is finite. The detailed experiment setup and references for these attack methods are presented in Section 4.1.
>
>
> Q.3 Policy of a given type
>
>
> Theoretically, the policy for a given attack type is a mapping from the global model weight $w$ to the gradient update $\tilde{g}$. Even though we do not include experiments under this generic setup, [3] demonstrate the effectiveness of this generic attack policy setup using synthetic data, see Figure 6 in Appendix E of that paper. In our experiments, the attack policy takes in the model weights and outputs hyperparameters of the underlying attack methods.
>
>
> Q.4 Dimension of $\theta$
>
>
> The defense action $a_D$ does live in $R^3$, yet Algorithm 1 aims for an optimal defense neural network policy, parameterized by $\theta$, which outputs the defense action. The dimension of $\theta$ depends on the network architecture.
>
> Q.5 Projection
>
> There is no dimension-wise projection. All outlier parameters in each coordinate are eliminated based on the fraction $b$.
>
>
> Q.7 Insights of OOD generalization
>
>
> We employ a coupling technique to analyze the generalization bound, which has been explored in the context of meta-supervised learning [1]. We extend this approach to the meta-RL scenario. The benefit of using this coupling technique is that the bound does not depend on the topology of the attack-type space. The message of this result is that the obtained pre-trained defense policy is still effective even if the attack method appearing in the online environment is not included in the pre-training.
>
> Q.8 Overparameterized model
>
> The defender's objective is optimized on clean data, which mitigates the backdoor effects automatically [2]. Our Reptile meta-learning (see Algorithm 2.3 in Appendix) structure accelerates this process, since it will calculate multi-step adaptation for each FL epoch based on clean data.
>
>
> Q.9 Violation of privacy and ablation
>
>
> Please refer to our discussion related to privacy concerns from the rebuttal to Reviewer YEKb's Q1 and the detail of the server generating simulated data in Appendix E. Without breaking the privacy requirement in FL, all the simulated data are generated using cGAN or Diffusion models. The quality of the generated model is influenced by many factors, e.g., root data, augmentation methods, and model training parameters. Once the generated models are trained, we can generate as much data as needed to simulate the FL environment. In our experiment, we generated 500 data for each client as in the real FL environment. To test the influence of data distribution shifts, we establish an ablation study in Appendix Figure 9(d), where the iid-levels of data are different between the simulated environment and the real FL environment.

---

> ### Author Response · Authors · 2023-11-20
> **Q.6 Convergence analysis and sample attack batch**
>
> The PL condition, as a customary assumption in the non-convex domain (see our rebuttal to Reviewer YEKb's Q5), can significantly simplify the approximation analysis for the max-oracle. However, consider quantifying the approximation error between the defender's gradient estimate under the attacker's last iterate $\nabla\_\theta \mathbb{E}\_{\xi \sim Q} \mathcal{L}\_D (\theta, \phi^t\_\xi (N_A), \xi)$, and the true gradient under the attacker's best response $\nabla\_\theta \mathbb{E}\_{\xi \sim Q} \mathcal{L}\_D(\theta, \phi^*(\xi), \xi)$. The challenge of analyzing the error can not be simply addressed by the typical approximation results of a max oracle in the literature [4], as we need to bridge from the attacker's gradient dynamics under $ \nabla\_\phi \mathcal{L}\_\mathcal{A}$  to the defender's value function $\mathcal{L}\_\mathcal{D}$, which was not yielded straightforwardly by strict competitiveness and other assumptions. Our technical contribution lies in analyzing the approximation error with the presence of gradient adaptation in $\mathcal{L}\_\mathcal{D}$.
>
> To the reviewer's second question, we indeed omitted the complexity analysis for the attack-type ($\xi$) sample size and the outer-loop batch size ($N_b$ for min-oracle) for the ease of exposition, which does not change the complexity results. We only stated the convergence result with respect to the error $\mathbb{E}[e_t]$ expected over actual attacker type prior $Q$, instead of noise policy gradients and empirical measure $\hat{Q}(\xi_i):= \frac{1}{K} \sum_{k =1}^K  \mathbb{I}_{\{\xi_k = \xi_i\}}$, $K:= |\hat{\Xi}|$.  $e_t$ is upper bounded using $z_t$, the discrepancy between the actual first-order information and the approximated one.
>
> Note that the analysis would not be so much different when we want to analyze the actual algorithm, we need to define
>     $$
>     \hat{z}\_t :=  \mathbb{E}\_{\xi \sim \hat{Q} }\hat{\nabla}\_{\theta}\mathcal{L}\_\mathcal{D} (\theta, \phi\_\xi^t(N_\mathcal{A}), \xi) -  \nabla\_{\theta} V(\theta),
>     $$ which account for the $\xi$ and defender gradients sampling, $\hat{z}\_t$ can be bounded by:
> $$
>     \| \hat{z}\_t \|  \leq \| z\_t \| + \Vert  \mathbb{E}\_{\xi \sim Q }\nabla\_{\theta}\mathcal{L}\_\mathcal{D} (\theta, \phi\_\xi^t(N\_\mathcal{A}), \xi) -  \mathbb{E}\_{\xi \sim \hat{Q} }\hat{\nabla}\_{\theta}\mathcal{L}\_\mathcal{D} (\theta, \phi\_\xi^t(N\_\mathcal{A}), \xi)\Vert
> $$
> $$
>      \leq \| z\_t \| +  \underbrace{\Vert  \mathbb{E}\_{\xi \sim Q }\nabla\_{\theta}\mathcal{L}\_\mathcal{D} (\theta, \phi\_\xi^t(N\_\mathcal{A}), \xi) -  \mathbb{E}\_{\xi \sim \hat{Q} }\nabla\_{\theta}\mathcal{L}\_\mathcal{D} (\theta, \phi\_\xi^t(N\_\mathcal{A}), \xi)\Vert }\_{I} + \underbrace{ \Vert \mathbb{E}\_{\xi \sim \hat{Q}} \left[ \nabla\_{\theta}\mathcal{L}\_\mathcal{D} (\theta, \phi\_\xi^t(N\_\mathcal{A}), \xi) -  \hat{\nabla}\_{\theta}\mathcal{L}\_\mathcal{D} (\theta, \phi\_\xi^t(N\_\mathcal{A}), \xi) \right] \Vert }\_{II}
> $$
> I and II can be viewed as additive noises. I is essentially the type sampling variance and can be controlled by making $K$ large enough; II is essentially the trajectory sampling variance and can be controlled by making $N\_b$ large enough, i.e.,
> \begin{align*}
>      I   \leq  \mathcal{O} (\frac{\sigma_\xi}{ \sqrt{K}})  \quad
>      II  \leq \mathcal{O} (\frac{\sigma}{\sqrt{N_b}}),
> \end{align*}
> where $\sigma_{\xi}$ is the policy gradient standard deviation of attack type distribution.
> Some calculations will still lead to $N_b \sim \mathcal{O}(\varepsilon^{-4})$ and  $K \geq \mathcal{O}(\varepsilon^{-4})$, and the construction of the constant coefficients will not be so much different, so the orders of the eventual results for $N_\mathcal{D}$, $N_b$, and $N_\mathcal{A}$ would remain the same.
>
> [1] Fallah, A., Mokhtari, A., \& Ozdaglar, A. (2021). Generalization of model-agnostic meta-learning algorithms: Recurring and unseen tasks. Advances in Neural Information Processing Systems, 34, 5469-5480.
>
> [2] Zhang, Zhengming, et al. "Neurotoxin: Durable backdoors in federated learning." International Conference on Machine Learning. PMLR, 2022.
>
> [3] Li, H., Sun, X. and Zheng, Z., Learning to attack federated learning: A model-based reinforcement learning attack framework. NeurIPS 2022.
>
> [4] Fallah, A., Georgiev, K., Mokhtari, A., \& Ozdaglar, A. (2021). On the convergence theory of debiased model-agnostic meta-reinforcement learning. Advances in Neural Information Processing Systems, 34, 3096-3107.

---

> > ### Comment · Reviewer_D75P · 2023-11-22
> >
> > I thank the authors for their rebuttal, but maintain my score. I suggest at least rewriting the paper to include the missing definitions/unclarified points mentioned.

---

### Official Review · Reviewer_YEKb · 2023-11-05

**Soundness:** 2 fair
**Presentation:** 3 good
**Contribution:** 2 fair
**Rating:** 3
**Confidence:** 5

**Summary:**

The authors develop Bayesian Stackelberg Markov game to handle security problems in FL and incomplete information.

 The authors have built on previous RL and meta learning literature to establish the game and solve the problem by constructing a simulation environment.

Though the problem is important, I found serious concerns with respect to the considered setting and its usefulness for FL, the developed solution, and results.

**Strengths:**

Addressing security issues in FL is an important problem. The problem is important, however, the idea is not original. The paper is relatively well written but the developed solution and contribution is not significant.

**Weaknesses:**

The main issue with this paper is the assumption for the existence of a pre-training. It is not clear which entity provides the data for pre-training as the data privacy is critical in FL. If the server does not have access to client's data, the pre-training step can be quite ineffective for example when the data distributions can be significantly different, which makes the overall proposed solution ineffective.

How do you generate the data for the simulation to ensure 1) the privacy is not violated 2) you make sure the distribution matches the distribution of data over honest clients?

--------

The sample complexity of the proposed method is exhaustive. The main problem is using an RL-based simulation methods, while there are alternative methods with guaranteed regret bounds with significantly smaller sample complexity. I am not sure whether an RL-based solution is a good idea to handle this problem.

--------

The setting of considering both backdoor attackers and untargeted attackers is not well motivated and counter intuitive. Since the objectives of those attackers are different, the overall attacks will be much less effective compared to considering two disjoint scenarios where you have either backdoor attackers or untargeted ones.

--------

The authors consider each attack separately and optimize policy according to each attack. It is quite time consuming. The setting assumes that the defender knows attacks distribution but it averages all adapted policies rather than does weight average.

--------

The Assumption 3.4. which is required for the following theoretical results are quite restrictive. Can the authors provide a concrete learning problem with deep neural networks that satisfy this assumption?

--------

Some relevant related work have not been discussed and compared.

Model-sharing games: Analyzing federated learning under voluntary participation. AAAI 2021.

Mixed gradient aggregation for robust learning against tailored attacks. TMLR 2022.

**Questions:**

How do you generate the data for the simulation to ensure 1) the privacy is not violated 2) you make sure the distribution matches the distribution of data over honest clients?


Can the authors provide a concrete learning problem with deep neural networks that satisfy Assumption 3.4?

---

> ### Author Response · Authors · 2023-11-20
>
> We would like to thank the reviewer for thoughtful and valuable comments on our paper.
>
> Q.1 Privacy concerns
>
> Please refer to the rebuttal to Reviewer gEMV's Q1 for the arguments related to the motivation of pre-training and online adaptation. We here focus on the privacy issue. We note that the pre-training does not require precise knowledge of clients' data or environmental parameters to simulate an environment. Our methodology allows distribution shifts between the simulated and actual FL environment, which does not invalidate the pre-trained defense policy. Our pre-trained policy consistently outperforms baselines as in Figure 9(c)(d) in Appendix F, despite variations in the iid-level of data and the fraction of malicious clients between pre-training and online actual FL.
>
> For our experiment specifically, although inference attacks (e.g., inverting gradient [1]) are applied to collect a limited amount ($<200$) of root data (which is unnecessary when the server possesses a small amount of root data as assumed in [2]), the exact original images are not fully reconstructed here due to the large batch size and the existence of attackers. We further add data augmentation (e.g., distortion, denoising, etc.) to not only enlarge the base data size, but also obfuscate the original data. Finally, we use those base data to train cGAN or Diffusion models to generate data for simulated environment. Such data can be extremely different from the original data either in the single data or distribution level. Please refer Appendix E for the details of self-generated data.
>
>
> Q.2 Efficiency of RL-based defense
>
>
> We first highlight that the overall complexity is of the same order as other meta-learning algorithms up to a logarithmic factor $\log\epsilon^{-1}$ due to the inner loop (the attack best response), as discussed in the first paragraph of Section 3. Compared with meta-learning algorithms, we don't consider our complexity exhaustive in theory. As discussed in the paragraph before section 3 and rebuttal to Reviewer gEMV's Q1, the simulated environment can be significantly smaller than the real FL without communication/transition overhead, ensuring the efficiency of RL. To improve efficiency, parallel training among clients and state/action space reduction are also applied to the simulated environment.
>
>
> It would be appreciated if the reviewer could offer the specific references mentioned in the comment. From our understanding, the reviewer refers to online learning algorithms (bandit algorithms) with regret bounds. However, modeling adversarial FL as a bandit problem fails to capture the dynamic process of FL. The global model at each step is jointly determined by the last attack actions, subsampling, and aggregation rules, affecting both players' future actions. Hence, it is more appropriate to model adversarial FL as a sequential decision-making problem, where the defender needs to care about the long-term future performance of the FL system at each time step. This sequential decision-making formulation directly lends itself to the RL approach.
>
>
> Another reason for choosing the RL approach is that some advanced adaptive attacks are learning-based as well, adjusting the attack actions to the FL process and targeting long-term attack effectiveness [3,4]. To combat these strong attacks, the defender needs to consider the attackers' learning process, leading to the proposed Stackelberg learning, emphasizing the long-term defense effectiveness when facing adaptive attacks. This long-term thinking in sequential decision-making would be absent if one chose the bandit formulation.
>
> Q.3 Mixing backdoor and untargeted attacks
>
>
> Figure 1(b) shows that multiple kinds of attacks can achieve their attack goals simultaneously, while a simple combination of traditional SOTA defenses against a single type of attacker fails to defend such a mixed attack. One of the major advantages of modeling FL security as a multi-agent game is to capture both the cooperative and competitive objectives among all agents, which leads to a successful defense against mixed types of attacks. In our attack objective modeling, unless the poison ratio (i.e., fraction of poisoning data) is tiny and the fraction of backdoor attackers is significantly larger than the fraction of untargeted attackers, both attacks will work simultaneously. In our experiment section, we evaluate cases where only one type of attack exists, which achieves the highest performance of their own goals.

---

> > ### Comment · Reviewer_YEKb · 2023-11-22
> > **After Rebuttal**
> >
> > I would like to thank the authors for their response. After reading the rebuttal carefully, I would like to keep my score unchanged mainly because I think the paper requires substantial revision to address privacy issues related to pre-training with some guarantees and also proper comparison with non RL-based alternatives in terms of robustness-computation trade-offs.

---

> > > ### Author Response · Authors · 2023-11-22
> > > **Related References**
> > >
> > > We thank the reviewer for the constructive comments. Could you please point us to some representative works with "guaranteed regret bounds with significantly smaller sample complexity?"
> > >
> > > Again, thank the reviewer for bringing up this line of work, which would help us to present our work better.

---

> ### Author Response · Authors · 2023-11-20
>
> Q.4 Optimize against each attack
>
>
> Meta-Stackelberg learning takes each sampled attack and derives a meta policy through optimization. However, the proposed learning does not tackle every possible attack from the distribution. By optimizing the meta policy against the finite sampled attacks, the resulting $\theta$ approximates the optimal solution to the expected objective function averaged over all attacks in Eq.(3) in the sense that $\theta$ generalizes well to attacks unseen in the pre-training stage.
>
> As for the reviewer's concern about efficiency, we stress that our algorithm design has delivered SOTA complexity, as discussed in Q.2. To speed up the learning process in RL simulation, we can resort to parallelization (e.g., multiprocessing), where sampling and gradient evaluation are operated concurrently for each sampled attack without interfering. For example, process-based parallelization has been widely adopted in meta-learning practice [5].
>
>
> Q.5 Practicality of Assumption 3.4
>
>
> This assumption extends the seminal condition in first-order methods in non-convex optimization [6]. [7] provides two classification examples where the PL condition is met. More closely relevant to our RL context, [8] empirically demonstrate that the cumulative rewards in meta-reinforcement learning satisfy the PL condition, see Figure 4 Appendix D in [8].
>
>
> [1] Geiping, Jonas, et al. "Inverting gradients-how easy is it to break privacy in federated learning?." NeurIPS 2020.
>
> [2] Cao, Xiaoyu, et al. "FLTrust: Byzantine-robust Federated Learning via Trust Bootstrapping." NDSS 2021.
>
> [3] Li, H., Sun, X. and Zheng, Z., Learning to attack federated learning: A model-based reinforcement learning attack framework. NeurIPS 2022.
>
> [4] Wen, Yuxin, et al. "Thinking two moves ahead: Anticipating other users improves backdoor attacks in federated learning.". ICML Frontiers Workshop (AdvML), 2022.
>
> [5] Finn, C., Abbeel, P., \& Levine, S. (2017, July). Model-agnostic meta-learning for fast adaptation of deep networks. In International conference on machine learning (pp. 1126-1135). PMLR.
>
> [6] Karimi, H., Nutini, J., \& Schmidt, M. (2016). Linear convergence of gradient and proximal-gradient methods under the polyak-łojasiewicz condition. In Machine Learning and Knowledge Discovery in Databases: European Conference, ECML PKDD 2016.
>
> [7] Nouiehed, M., Sanjabi, M., Huang, T., Lee, J. D., \& Razaviyayn, M. (2019). Solving a class of non-convex min-max games using iterative first-order methods. Advances in Neural Information Processing Systems, 32.
>
> [8] Li, T., Lei, H., \& Zhu, Q. (2022). Sampling Attacks on Meta Reinforcement Learning: A Minimax Formulation and Complexity Analysis. arXiv preprint arXiv:2208.00081.

---

### Official Review · Reviewer_gEMV · 2023-11-05

**Soundness:** 3 good
**Presentation:** 3 good
**Contribution:** 3 good
**Rating:** 5
**Confidence:** 3

**Summary:**

This paper proposes a new federated learning defense mechanism by combining Bayesian Stackelberg Markov game and meta-learning.
The proposed procedure contains a pre-training stage that learns a meta defend policy in simulated environment, and an online execution stage, where the meta defend policy is updated using data collected from real interactions with the potentially malicious clients.

**Strengths:**

The idea of using Bayesian Stackelberg Markov game to model adversarial federated learning seems to be novel.

**Weaknesses:**

Please correct me if I am missing anything.

In the proposed defense framework, we will train three neural networks, one for minimizing the FL loss, with parameter $w$, one for the policy network of the defender, with parameter $\theta$, and one for attacker (could be multiple actually, we have one for each type of attacker), with parameter $\phi$. Then when we apply this in real FL environment, we rely on the defender's policy network to alter the updated global model at each iteration, to defende against potential attackers. And importantly, the learned defender's policy will only work for the particular neural network used for minimizing FL loss during pre-training, i.e., if we change the dataset (e.g., change in distribution of x, or conditional distribution of y|x), choose a different loss function, or change the structure of the neural network, the defender policy no longer works. Is this correct?

What I am a little bit confused about is, what this framework seems to be saying is that, we have already have enough data to train a neural network $w$ during pre-training to solve some FL task. Now what I am worried about is that, if I train this neural network again in real FL environment (ideally, using exactly the same training data as the one used during pre-training), there might be some attackers that manipulate my global model updates. Therefore, to defend against them, I will create a simulation environment that contains different type of attackers, and I train my neural network $w$ to solve the FL task, with the help of a defender policy $\theta$ that tries to "correct" the manipulated global model. Now with the trained defender policy $\theta$ (with some online adaptation), I can properly train my network $w$ in real FL environment in the face of potential attackers.

If this is true, then it seems FL is not really necessary, since we are already capable of training a good $w$ locally? I'd appreciate it if the authors can shed more light on this.

**Questions:**

1. There seems to be a bit mismatch in the description of how the defense strategy works.

In Section 2.1 paragraph "FL process", the server applies a post-training defense $h(\cdot)$ only on the final global model, i.e., $h(w_{g}^{T})$. In Section 2.2, the action of the defender is described as $a_{D}^{t}=h(w_{g}^{t+1})$, which is applied at each step.

2. The definition of defender’s expected utility in Section 2.3

Intutively, we should only care about the final loss of the FL procedure, i.e., how good our model is at the end of FL. But the currently definition says we should care about the cumulative loss during the whole FL procedure, which does not seem to be very reasonable? Perhaps changing the reward definition in Section 2.2 to "reduction of loss compared with last iteration", instead of loss at current iteration would be better?

3. I'd appreciate if the authors can elaborate on what it would mean if the proposed method successfully defend against the attackers, e.g., in terms of convergence rate of the FL procedure. This seems to be vague in the current presentation of the theoretical results.

---

> ### Author Response · Authors · 2023-11-20
>
> We appreciate the reviewer's insightful comments.
>
> Q.1 Defense generalization and necessity of real FL
>
> The reviewer's question relates to the fundamental motivation of the pre-training and then adaptation approach. In our proposed method, the server gains only rough insight into the general data distribution and potential attack methods without delving into the specifics of individual clients or the broader environment. This strategic approach safeguards privacy in Federated Learning (FL) by avoiding the need to access detailed client/environment information.
>
> Typically, the pre-training phase occurs in a simulated environment, which is relatively small (consisting of, say, 10-100 clients). This compact setting enables the efficient training of an initial defense policy. In contrast, the actual FL environment may be considerably more intricate and slower, involving over 1000 clients with associated communication overheads.
>
> The FL model learned in the simulated environment cannot be directly applied due to shifts in data distribution and uncertainties in security threats. However, the defense policy (embodied as a neural network in our context) acquired from this small and approximative simulated environment proves effective after limited adaptation in the real environment, as supported by both of our theoretical reasoning and empirical evidence.
>
> The pre-training environment setup targets the specific FL task, where the neural network's loss function and structure are already determined and known to the defender. Even though we agree with the reviewer that the learned defense policy may not be effective when facing different loss functions and network structures in the online adaptation phase, we want to emphasize that our proposed meta-SG framework mainly targets the defender's incomplete information on the uncertain and unseen attacks rather than on the FL task.
>
> Q.2 Mismatch in defense strategy description
>
> This is not a mismatch but a deliberate design. The mapping $h$ applied to each step is not for post-training defense but for evaluating its effectiveness. The output of this mapping $\hat{w}_g^t=h(w_g^t)$ does not influence the FL process. The actual global model weights are $w_g^t$, and $\hat{w}_g^t$ is merely for the reward computation.
>
> Q.3 Definition of Defender's utility
>
> The success criterion of the defense is measured through the average accuracy of the global models, which naturally leads to the cumulative loss during the whole FL procedure. The limitation of using the final loss is that clients will join and leave the FL system during the process, of which the server is unaware. Hence, we consider a long horizon and apply the discounting factor in the pre-training to encourage arriving at high accuracy as soon as possible. If one were to use the "reduction of loss compared with the last iteration," i.e., $-E[F(\hat{w}_g^{t+1})]+E[F(\hat{w}_g^{t})]$, then the cumulative rewards would be $-\gamma^H E[F(\hat{w}_g^{T})]$ for some large $H$. Consequently, the coefficient $\gamma^H$ is extremely small, rendering the loss insignificant.
>
> Q.4 Success criterion and interpretation of convergence
>
> The success criterion concerns the average accuracy of the global models in the real FL environment, while the convergence rate and associated theoretical analysis concern the pre-training efficiency. These theoretical results confirm that the proposed pre-training algorithm (Alg. 1) retains the SOTA complexity and sample efficiency, which delivers the finalized defense policy as quickly as other meta-learning algorithms.

---

### Meta-Review · Area_Chair_6x5J · 2023-12-12

**Metareview:**

This paper addresses security vulnerabilities in federated learning (FL) systems by formulating adversarial scenarios as a Bayesian Stackelberg Markov game (BSMG). Introducing an efficient meta-learning approach, the proposed algorithm proves to converge to the first-order equilibrium point in a theoretically sound manner. Empirical results highlight its robustness against adaptive attacks. While the problem is interesting most of the reviewers unanimously agree that the paper requires significant revision and clarification on the results / missing claims.

We encourage the authors to submit to the next suitable venue after making the changes.

**Justification For Why Not Higher Score:**

N/A

**Justification For Why Not Lower Score:**

N/A

---

### Decision · Program_Chairs · 2024-01-16

Reject